# How seasonal hydroclimate variability drives the triple oxygen and hydrogen isotope composition of small lake systems in semiarid environments

Claudia Voigt[1,2], Fernando Gázquez[1,2], Lucía Martegani[1,2], Ana Isabel Sánchez Villanueva[1,2], Antonio Medina[3], Rosario Jiménez-Espinosa[3], Juan Jiménez-Millán[3], and Miguel Rodríguez-Rodríguez[4]

[1]Department of Biology and Geology, University of Almería, Almería, 04120, Spain
[2]Analusian Center for Global Change – Hermelindo Castro (ENGLOBA), University of Almería, 04120, Spain.
[3]Department of Geology and Center for Advanced Studies in Earth Sciences, Energy and Environment (CEACTEMA), University of Jaén, Jaén, 23071, Spain
[4]Department of Physical, Chemical and Natural Systems, University Pablo de Olavide, Seville, 41013, Spain

*Correspondence to*: Claudia Voigt (cvoigt@ual.es)

**Abstract.**

This research investigates the influence of seasonal hydroclimate variability on the triple oxygen and hydrogen isotope composition of small, shallow lake systems that show substantial intra- and interannual fluctuations in the water level. The study was conducted at Laguna Honda, a semi-permanent lake located in the semiarid Mediterranean environment of southern Spain. Over one year, lake water level was monitored continuously and water samples from the northern and southern margin were taken monthly for major ion concentration and triple oxygen and hydrogen isotope analyses. Over the study period, the lake water level dropped from 1.4 m to 0.6 m, while salinity increased from 23 g L$^{-1}$ to 130 g L$^{-1}$ and $\delta^{18}O$, $\delta^2H$ and $^{17}O$-excess of lake water varied from -2 ‰ to 15 ‰, -26 ‰ to 51 ‰ and -9 per meg to -87 per meg, respectively. Hydrological mass balance calculations indicate that precipitation, basin discharge and evaporation control lake water level changes in Laguna Honda, and major inflow from other sources, such as groundwater, is absent. The lake water's isotope composition is mainly driven by changes in relative humidity (34-73 %), while precipitation and basin discharge can cause transitional mixing effects that however remain small in magnitude (< 10 %). In the $^{17}O$-excess vs. $\delta'^{18}O$ space, the lake water forms a loop evolving from low $\delta^{18}O$ and high $^{17}O$-excess in winter to higher $\delta^{18}O$ and lower $^{17}O$-excess in summer along a convex curvature, and back to low $\delta^{18}O$ and high $^{17}O$-excess with the beginning of the subsequent rainy season along a concave curvature. The triple oxygen isotope system allows the identification of non-steady state conditions, which is challenging using $\delta^2H$ and $\delta^{18}O$ alone, due to the linearity of trends in this isotope system. The large seasonal variability of triple oxygen isotopes should be considered when interpreting isotope data obtained from paleo-archives from lake sediments in semiarid and arid environments.

## 1 Introduction

Stable isotopes of water ($\delta^2H$, $\delta^{18}O$) provide a powerful tool for hydrological balancing of lacustrine systems (Gat, 1995; Gibson et al., 2016b; Krabbenhoft et al., 1990). Such isotopic studies often trace the

mean annual water balance of large-volume lakes, using basic models that assume isotopic and hydrological steady state (Gibson et al., 2016a; Jasechko et al., 2014; Zanazzi et al., 2020; Zuber, 1983). However, large intra- and interannual fluctuations in the water level of small lacustrine systems in semiarid environments induced by the seasonality of precipitation and climate challenge these methods

to accurately assess the lake's water balance and force the application of more complex non-steady state models (Gibson, 2002). Triple oxygen isotopes are a novel isotope tracer that hold potential to improve the understanding of hydrological functioning of such highly dynamic lake systems. The secondarily-derived parameter $^{17}$O-excess [= $\delta'^{17}O - 0.528\,\delta'^{18}O$ with $\delta' = 1000\ln(\delta/1000+1)$] is only weakly influenced by temperature changes and highly sensitive to kinetic isotope fractionation occurring during

evaporation (Barkan and Luz, 2005; Luz et al., 2009). Further, in triple oxygen isotope space ($^{17}$O-excess over $\delta'^{18}O$), fractionation and mixing processes form curves (Herwartz, 2021), not lines as in the traditional $\delta^2$H-$\delta^{18}$O system. Recent studies showed that this curved behaviour of the triple oxygen isotope composition of lake water undergoing evaporation can serve to determine the connectivity of the lake to underlying groundwater aquifers (Surma et al., 2015, 2018) and to identify admixture of freshwater

sources, e.g., during occasional precipitation events (Voigt et al., 2021). Lately, Pierchala et al. (2021) demonstrated the applicability of $\delta^{17}$O for lake hydrological balancing of a groundwater-recharged permanent lake in a mid-latitude environment. However, the response of lake's triple oxygen isotope composition to seasonally changing environmental conditions has not been investigated so far.

The developments in the field of triple oxygen isotopes of lake water have also led to first applications to

paleoclimate records. Recent studies on $^{17}$O-excess of lake carbonates revealed their potential to provide information on catchment precipitation and primary water sources (Passey and Ji, 2019; Passey and Levin, 2021). Gázquez et al. (2018) and (2023) demonstrated to utility of combining the analysis of triple oxygen and hydrogen isotopes of gypsum hydration water extracted from lacustrine gypsum to quantitatively estimate past changes in relative humidity. An accurate interpretation of these paleo-archives requires a

fundamental understanding of processes driving the variability of $^{17}$O-excess in hydrologically different lake systems and their dynamics.

In this study, we investigate how seasonal changes in environmental variables drive triple oxygen and hydrogen isotopes of Laguna Honda, a semi-permanent lake located in the semiarid Mediterranean environment of southern Spain. The Andalusian lowlands contain an abundance of coastal and inland

wetlands that provide a critical habitat for a diverse array of plants and animals (Rodríguez-Rodríguez, 2007). However, increasing water demand due to population growth, urbanization and economic development threaten these ecosystems. The Andalusian lowland lakes are particularly vulnerable, as low rainfall and high evaporation already limit water availability, and impermeable materials impede groundwater discharge in many of these lakes (Rodríguez-Rodríguez, 2017; Rodríguez-Rodríguez et al.,

2010). Anthropogenic climate change will further intensify the water deficit problem. Climate models predict an increase in the frequency, intensity, duration and extent of hot extremes and agricultural and ecological droughts over the next century in the Mediterranean region (IPCC, 2023). Understanding the hydrological functioning of lake systems is essential to preserve these ecosystems and maintain biodiversity.

In January 2021, a hydrological station was installed in the southern part of Laguna Honda to continuously monitor lake water level changes. Further, precipitation and lake water samples were collected monthly over one year for isotopic ($\delta^{18}$O, $\delta^2$H and $^{17}$O-excess) and chemical analyses. Daily lake water level

changes were simulated using meteorological data obtained from a nearby weather station and a previously published lake bathymetric map. Simulation results were compared to observations to identify the major hydrological processes controlling the lake water balance. An isotope mass balance model accounting for lake volume changes was used to simulate the evolution of triple oxygen and hydrogen isotopes of the lake water over the study period. Sensitivity experiments were performed to evaluate the robustness of the model results and identify the environmental variables that drive isotope variations of lake water. The results provide insights into the hydrological functioning of the lake and its future development as well as the usefulness of triple oxygen isotopes for hydrological balancing.

## 2 Site description

Laguna Honda (460 m a.s.l., 37°35'52.7"N 4°08'34.5"W) is a saline to hypersaline lake (up to 130 g L$^{-1}$) of about 0.085 km$^2$ extension located in an endorheic basin in southern Spain (Fig. 1; Rodríguez-Rodríguez et al., 2010). It is formed by a karstic depression fed mainly by seasonal precipitation and surface runoff. The lake basin has an oval morphology, with the major axis running from north to south. The deepest area lies in the southern part of the lake, reaching a maximum depth of 3.16 m (Castro et al., 2003). The banks slope gently to the eastern and northern margin, while steeper slopes stand out in the south (Fig. 1b). The catchment area of approximately 0.96 km$^2$ has an undulating topography with a marked drainage in east-west direction (Rodríguez-Rodríguez et al., 2010). It is cultivated with olive groves, which extend almost to the edge of the lake's southern margin. The geological substrate mainly consists of Triassic marls and clays, olistoliths of gypsum and isolated blocks of carbonate rocks (Medina-Ruiz et al., 2024). The low permeability of the Triassic material inhibits the formation of larger groundwater aquifers and results in basin drainage occurring mainly on the surface.

The regional climate is characterized by semi-arid to sub-humid conditions, marked by a dry season lasting from June to September and a wet season from October to May, with an average annual rainfall of 490 mm (2001-2023; Junta de Andalucía, 2024). The long-term average temperature and relative humidity vary from 8 °C and 72 % in winter to 27 °C and 34 % in summer, respectively (Junta de Andalucía, 2024). The seasonal climate along with the basin's hydrogeology make the lake a semi-permanent system, experiencing a period of summer drying that varies according to the seasonally occurring rainfall, with evaporation clearly dominating over precipitation. The seasonality and large interannual variability in the frequency and intensity of rainfall cause notable fluctuations of Laguna Honda's water level, as observed in many wetlands in southern Spain (Jiménez-Bonilla et al., 2023; Rodríguez-Rodríguez et al., 2010).

## 3 Materials & Methods

### 3.1 Meteorological data and morphometry

Daily rainfall amount, atmospheric temperature, relative humidity and potential evapotranspiration were continuously recorded at the meteorological station *Alcaudete* located 6 km east of Laguna Honda (Junta de Andalucía, 2024). Potential evapotranspiration is assumed to be equal to evaporation from the lake

surface. Morphometric data were obtained from a previous study on lake bathymetry (Castro et al., 2003). These data allowed the quantification of the flooded surface area at certain lake water levels using the Free and Open Source Software QGIS, which were then linearly interpolated (Fig. 1c). The lake volume was estimated based on the surface area of slices at discrete depth intervals of 0.01 m (Fig. 1d).

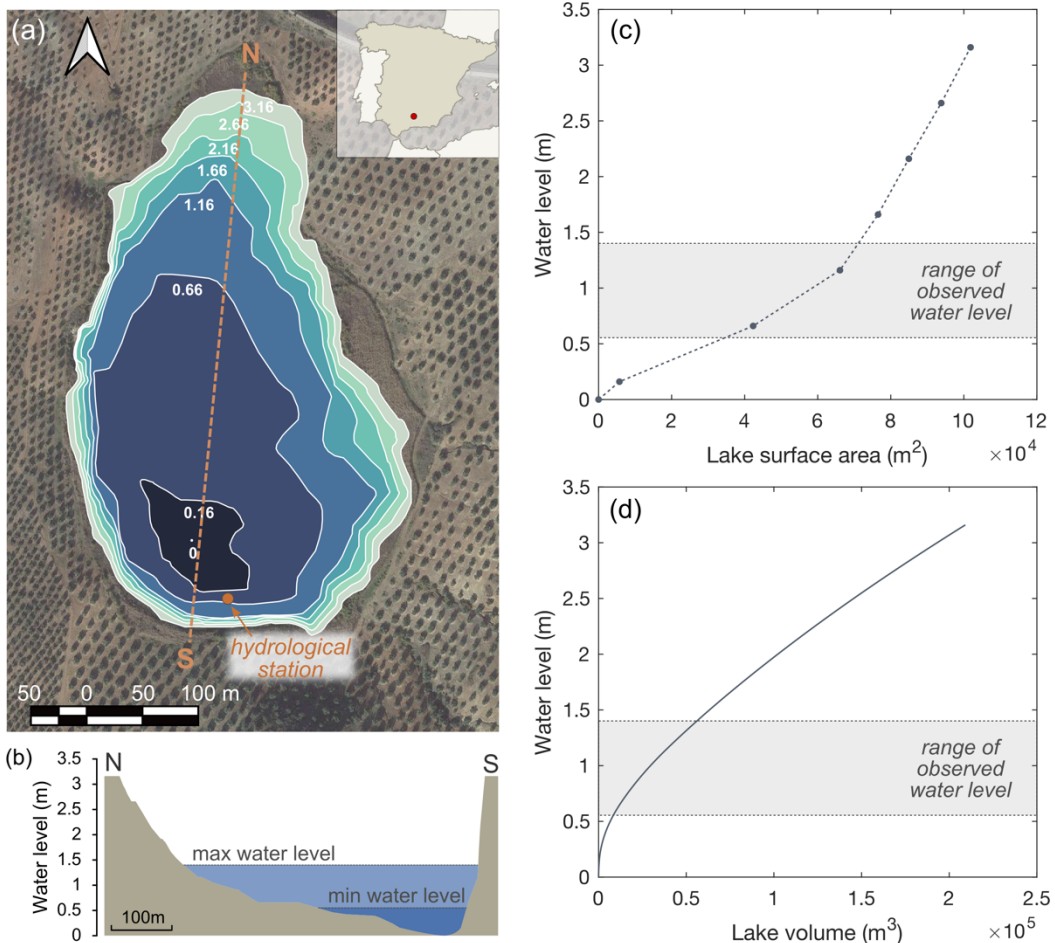

Figure 1: (a) Bathymetric map of Laguna Honda after Castro et al. (2003). The small numbers along the contour lines indicate the water level in (m). The brown circle marks the location of the hydrological station. (b) Elevation profile along the N–S line indicated in (a). The minimum and maximum water level observed over the study period are indicated. (c) and (d) Relationships between water level, lake surface area and lake volume. Lake surface area was determined at certain lake water levels (dots) and linearly interpolated (dashed line). The lake volume was estimated based on the surface area of slices at discrete depth intervals of 0.01 m.

### 3.2 Hydrological monitoring

From February 2021 to June 2022 a hydrological station was installed in the southern part of the lake (Fig. 1a). Absolute pressure (atmospheric pressure and water head) and lake water temperature were measured in 4-hour intervals using a HOBO U20L Water Level Logger (Onset Computer Corporation, Bourne, MA, USA) installed close to the lake bottom in a PVC pipe with holes to allow water to enter but protect the sensor from external agents. A second HOBO U20L Water Level Logger was placed in

the top of the pipe for measurements of atmospheric pressure and air temperature. During data processing, these data were used to compensate the absolute pressure data for atmospheric pressure variations. The compensated absolute pressure was then converted to lake water level readings and averaged to daily values. The obtained water level readings represent the water column above the sensor but underestimate the actual lake water level as the hydrological station is not situated at the deepest point of the lake. To

estimate the depth of the sensor with respect to the deepest point of the lake, lake surface area was estimated from Copernicus Sentinel-2 satellite imagery at biweekly to monthly time intervals over the study period using the Free and Open Source software QGIS. Comparison of the water level-to-lake surface area relationship obtained from these satellite data to that obtained from lake bathymetry (see Fig. 1c) revealed that the observed water level underestimates the actual water level by about 0.5 m (Fig. A1).

Lake water level data presented in this study have been corrected for this offset.

### 3.3 Isotope data and water chemistry

Rainwater was collected in a custom-made system consisting of a plastic bottle filled with a 1 cm thick oil film to prevent evaporation and topped by a funnel. The precipitation collector was sampled monthly from January 2021 to January 2022. Over the same period, lake water samples were collected monthly at

the southern and northern lake margin. As the water level dropped, the northern margin of the lake was no longer accessible from August 2021, so no further samples were taken at this location.

The isotope composition of rain and lake water samples was determined by a Picarro L2140-i cavity ring down spectrometer (Picarro Inc., Santa Clara, CA, USA) at the Laboratory of Stable Isotopes of the University of Almería, Spain. The measurement protocol is described in detail in Gázquez et al. (2022).

In brief, an autosampler (A0325, Picarro Inc., Santa Clara, CA, USA) was used to inject each sample ten times into a vaporizer (A0211, Picarro, Inc., Santa Clara, CA, USA), which was heated to 110 °C. Before introduction of the sample gas into the analyser, the sample was guided through the Picarro A0214 micro-combustion module (Picarro Inc., Santa Clara, CA, USA) to remove potential contamination produced by dissolved organic compounds (Gázquez et al., 2015). The first three injections were discarded to account

for memory effects from previous samples. The values of the remaining 7 injections were averaged. The isotope data were normalized to the VSMOW-VSLAP scale using four internal water standards analysed bracketing each set of 15 to 20 samples. The long-term precision for $\delta^{17}O$, $\delta^{18}O$, and $\delta^2H$ was 0.08 ‰, 0.15 ‰, and 0.7 ‰, respectively, based on the analysis of an analytical standard (n=35) conducted alongside the samples during the analysis period (April 2021 to April 2022). For $^{17}O$-excess and d-excess,

the precision was 13 per meg and 1 ‰, respectively.

Complementary ion concentration data of cations ($Na^+$, $K^+$, $NH_4^+$, $Ca^{2+}$, $Mg^{2+}$) and anions ($Cl^-$, $F^-$, $Br^-$, $NO_3^-$, $SO_4^{2-}$) were determined by ion chromatography using an Ion Chromatograph Professional IC 850–919 IC Autosampler Plus (Metrohm AG, St. Gallen, Switzerland) at the Department of Geology of the University of Jaén, Spain. Further, saturation indices of several minerals (calcite, aragonite, dolomite,

gypsum, anhydrite and halite) have been calculated for lake water samples using the Aquachem®10.0 software (Waterloo Hydrogeologic) following Parkhurst and Appelo (1999) and Drever (1997). The saturation index is calculated based on: SI = log (IAP/ KT) where IAP is the ionic activity product and KT the equilibrium at constant temperature for each mineral. SI values higher than zero indicate that

mineral precipitation is likely, while SI values lower than zero suggest that mineral dissolution can take place.

### 3.4 Mass balance simulations

Three mass balance approaches haven been applied in this study. First, hydrological data are used to simulate lake water level changes over the study period. In a second approach, lake water level changes are simulated using chemical data to assess their accuracy for lake water level reconstruction. Finally, hydrological and isotopic data are combined in an isotope mass balance model to simulate the daily evolution of triple oxygen and hydrogen isotopes in lake water over the study period. Sensitivity experiments are performed to identify the environmental factors that drive lake's isotope composition. The three approaches are outlined in detail below.

#### 3.4.1 Hydrological mass balance model

The change in lake water volume ΔV of Laguna Honda over a time step Δt is determined by:

$$\frac{\Delta V}{\Delta t} = V_{t+1} - V_t = P_{SA} + BD - E_{SA}, \tag{Eq. 1}$$

where $P_{SA}$ is water input by precipitation on the lake surface, $BD$ is water input by basin discharge, and $E_{SA}$ is water output by evaporation from the lake surface. Previous studies have shown that Laguna Honda is disconnected from the groundwater aquifer (Moral et al., 2008), which is therefore not accounted for in the equation. The initial lake volume $V_0$ is derived from lake bathymetry (Fig. 1d). Subsequent changes in the lake volume are simulated on a daily time step. For each time step, the lake surface area is estimated from the simulated lake volume $V_t$ using the surface area-to-volume ratio ($SA_t/V_t$) obtained from lake bathymetric data (Fig. 1c and d). Daily precipitation and potential evapotranspiration are obtained from the nearby meteorological station *Alcaudete* (Junta de Andalucía, 2024) and then multiplied by the lake surface area. $BD$ is estimated using daily soil water balances, calculated using the TRASERO software (TRASERO, 2015). $BD$ was simulated on a daily time step from 01 January 2020 to 31 July 2022. The simulation of $BD$ was started one year before the study period to account for potential pre-wetting of the soil. The simulated $BD$ was multiplied by the watershed area (0.96 km2; Rodríguez-Rodríguez et al., 2010) to obtain the total volume of $BD$ discharged onto the lake from the drainage basin. The $BD$ depends greatly on the water holding capacity (WHC) of the soil (Fig. A2). We adjusted the WHC to 190 mm, for which the simulated and observed lake volume and water level change are in close agreement (Fig. 2). Similar or even higher values of WHC have been used in previous studies on Andalusian lakes with similar basin geology (Jiménez-Bonilla et al., 2023; Rodríguez-Rodríguez et al., 2015). Note that the total amount of $BD$ increases when using lower WHC, but in all modelled scenarios $BD$ remains restricted to short periods of heavy or frequent rain events between February and April (Fig. A2).

#### 3.4.2 Salt mass balance model

Changes in lake volume can also be estimated from salt mass balance:

$$\Delta V = V_{t+1}c_{L,t+1} - V_t c_{L,t} = P_{SA}c_P + BD c_{BD} - E_{SA}c_E, \tag{Eq. 2}$$

where $c$ is the ion concentration (in mol m$^{-3}$) in precipitation ($P$), basin discharge ($BD$), evaporation ($E$) and lake water ($L$), respectively. We calculated the salt mass balance based on the concentration of chloride that is usually assumed to be conservative due to the high solubility of chloride-bearing minerals. Assuming that the chloride concentration in precipitation and basin discharge is negligible compared the lake water, the volume at time step $t+1$ is given by:

$$V_{t+1} = V_t \frac{c_{L,t}}{c_{L,t+1}},$$ (Eq. 3)

This assumption is supported by measurements of rainwater samples that reveal concentrations of chloride in rainwater that are usually 0.01-0.05 g L$^{-1}$ and always lower than 0.11 g L$^{-1}$, which is 3-4 magnitudes lower than that of lake water (23-130 g L$^{-1}$). Using other conservative ions (e.g. potassium or bromide) to calculate the salt mass balance does not significantly influence the results. The obtained lake volume is converted to water level using the lake volume-to-water level relationship given in Figure 1d.

### 3.4.3 Isotope mass balance model

The isotope mass balance of Laguna Honda is determined by:

$$\Delta V = V_{i+1}\delta_{L,i+1} - V_i\delta_{L,i} = P_{SA}\delta_P + BD\delta_{BD} - E_{SA}\delta_E,$$ (Eq. 4)

where $\delta_P$, $\delta_{BD}$ and $\delta_E$ are the isotope values of precipitation, basin discharge and the evaporation flux, respectively. $\delta_P$ is obtained from monthly collected rainwater samples. $BD$ occurred only in the rainy season, when evaporation is limited, within a few days after heavy rainfall events. Therefore, $\delta_{BD}$ is assumed to be equal to precipitation. $\delta_E$ is estimated from the Craig and Gordon isotope evaporation model (C-G model; Craig and Gordon, 1965; Criss, 1999):

$$R_E = \frac{R_L - \alpha_{eq}hR_V}{\alpha_{eq}\alpha_{diff}^n(1-h)},$$ (Eq. 5)

and $\delta = R - 1$. $R_L$ and $R_V$ are the isotope ratios of lake water and atmospheric water vapor, respectively, $h$ denotes the relative humidity normalized to surface water temperature and $\alpha_{eq}$ and $\alpha_{diff}$ are the isotope fractionation factors for equilibrium and diffusive fractionation, respectively. For $^{18}O/^{16}O$ and $^2H/^1H$, equilibrium fractionation factors from Horita and Wesolowski (1994) and diffusive fractionation factors from Majoube (1971) were used. The fractionation factor for $^{17}O/^{16}O$ is obtained from $^{17}\alpha = {}^{18}\alpha^\theta$ using $\theta_{eq}$ of 0.529 for equilibrium fractionation (Barkan and Luz, 2005) and $\theta_{diff}$ of 0.518 for diffusive fractionation (Barkan and Luz, 2007). The diffusive fractionation factor is modified by the empirical turbulence coefficient $n$ that accounts for the fraction of molecular diffusion of molecules on total kinetic isotope fractionation during evaporation, influenced by the wind conditions. In accordance with previous studies, $n$ was set to 0.5 (Gonfiantini, 1986). The $\delta^{17}O$, $\delta^{18}O$ and $\delta^2H$ of atmospheric water vapor are assumed to be in isotope equilibrium with precipitation: $\delta_V = (1000 + \delta_P)/\alpha_{eq} - 1000$. For periods in summer, when no precipitation occurred, the average isotope value of precipitation samples collected before and after the dry period was used to estimate the equilibrium water vapor values.

Note that continuous evaporation drives the isotope composition of the lake water towards an isotope steady state value (Criss, 1999):

$$R_{SS} = \frac{h\alpha_{eq}R_V}{1 - \alpha_{eq}\alpha_{diff}^n(1-h)}$$ (Eq. 6)

The isotope composition of this steady state value (also known as isotopic end value) is thus controlled by the relative humidity, the isotope composition of atmospheric water vapor and the turbulence

coefficient, and independent from the isotope composition of the lake water. Integrating Eq. 6 into Eq. 5 yields:


$$R_E = \frac{R_L - R_{SS}}{\alpha_{eq}\alpha_{diff}^n(1-h)} - R_{SS} \qquad \text{(Eq. 7)}$$

This equation shows that, if the lake water reaches isotope steady state ($R_L = R_{SS}$), the isotope composition of the evaporation flux will be equal to that of the isotope steady state value (and thus the lake water).

### 3.5 Sensitivity experiments

The isotope composition of the lake water is mainly controlled by i) the flux rates of evaporation and
precipitation that are controlled by the lake surface area and its ratio to the lake volume, and ii) the isotope composition of the evaporation flux that depends mainly on relative humidity, the isotope composition of water vapor in the ambient atmosphere and the turbulence coefficient. We modelled the isotope mass balance varying each of these input parameters, keeping other parameters constant, to evaluate their impacts on the model result.

### 3.5.1 Precipitation and evaporation fluxes

Both the fluxes of precipitation and evaporation are driven by the lake surface area, while their impact on the lake water isotope composition depends on the ratio of the surface area relative to the lake volume. The surface area-to-volume ratio can vary significantly, and often increases as the lake volume decreases, as for Laguna Honda. We assessed the influence of these parameters by varying the initial lake water
level by ±0.1 m, affecting the initial lake volume and thus the associated surface area. Further, we varied the surface area-to-volume ratio by ±0.1 to assess the influence of its uncertainty on the simulated isotope composition of lake water.

### 3.5.2 Isotope composition of the evaporation flux

The isotope composition of the evaporation flux is not measured but estimated from the C-G model (Eq.
5; Craig and Gordon, 1965). According to this model, ambient relative humidity, the isotope composition of water vapor in the ambient atmosphere and the turbulence coefficient are the major drivers of the isotope composition of the evaporation flux. It is further modified by the isotope difference between lake water and the theoretical isotope steady state value (see Eq. 7). The sensitivity of the isotope composition of the steady state value to each of these parameters and its impact on the isotope composition of lake
water has been assessed varying each of the parameters in their uncertainty margins described below. Daily average relative humidity data are obtained from a nearby meteorological station, but actual relative humidity at the lake site may deviate from these observations. Lakes often create their own microclimate, modifying temperature, relative humidity and wind patterns. Evaporation of lake water may lead to moisture buildup in the atmosphere and increase the relative humidity above the lake (Benson and White,
1994; Gibson et al., 2016b). On the other hand, the effective relative humidity during evaporation may be lower than its daily average value as the evaporation rate is usually higher during daytime when relative humidity is lower (Gibson, 2002). Also, temperature differences between lake water and the atmosphere may influence evaporation rates and effective relative humidity. Constraining the effective relative

humidity that drives the isotope composition of the evaporation flux with precision is thus challenging.
Here, we assess the impact of ±5% deviations from the effective relative humidity on the isotope composition of the lake water.

The isotope composition of atmospheric water vapor is not measured but estimated from isotope equilibrium with precipitation. However, precipitation occurs seasonally and only sporadic at our study site so that the equilibrium assumption may not always be valid. Further, rain drop re-evaporation
(Giménez et al., 2021; Landais et al., 2010) or convective activity (He et al., 2021; Landais et al., 2012), can decouple the precipitation from the atmospheric water vapor isotope signal, and especially impact the estimate of the [17]O-excess in water vapor. Here, we assess the impact of ±25% (0.25) deviations from equilibrium assumption ($V_{eq}$) on the isotope composition of the lake water. Further, we explore how the [17]O-excess of lake water evolves when keeping [17]O-excess of atmospheric water vapor constant (22 per
meg).

The atmospheric turbulence is determined by the fraction of molecular versus turbulent diffusion of molecules during evaporation. While molecular diffusion leads to kinetic isotope fractionation, no isotope fractionation occurs during turbulent diffusion. This is accounted for by the turbulence coefficient that is added as an exponent to the kinetic fractionation factor. The turbulence coefficient theoretically varies
from 0 (turbulent diffusion) to 1 (molecular diffusion), but a value of 0.5 has been found to best reflect natural conditions during lake evaporation (Gonfiantini, 1986). Previous studies on Spanish lakes used slightly higher values between 0.55 and 0.65 (Gázquez et al., 2018, 2023). Other studies found that a turbulence coefficient lower than 0.5 better explains the observed isotope data and relate this to moisture buildup in the atmosphere (Gat, 1995; Vallet-Coulomb et al., 2006). Here, we use a turbulence coefficient
of 0.5 and assume an uncertainty margin of ±0.1 to assess the impact of potential seasonal variations.

## 4 Results

### 4.1 Meteorological data

From February 2021 to July 2022, a total precipitation amount of 595 mm was recorded at the meteorological station *Alcaudete*. About one third of it occurred from February to June 2021, while the
other two thirds occurred between September 2021 and May 2022. No precipitation occurred in July and August 2021 or June 2022. Over the same period, potential evapotranspiration exceeded precipitation almost three times, cumulating to 1727 mm. Daily potential evapotranspiration rates showed significant seasonality ranging from ~1.5 mm per day in winter to ~6.2 mm per day in summer. Compared to long-term average conditions (2001-2023), annual precipitation was significantly lower (~15%) in the study
period, principally resulting from a shorter rainy season. In contrast, annual potential evapotranspiration and monthly averages of temperature and relative humidity agreed with long-term observations. Temperature ranged from 9 ºC in January 2021 to 27 ºC in July and August 2021. Relative humidity varied inversely from 77 % in January 2021 to 36 % in July 2021. Daily wind speeds ranged between 0.2 and 2 m s$^{-1}$ with windy days (daily average wind speed > 4 m s-1; Obermann et al., 2018) occurring more
frequently in summer.

**4.2 Hydrological mass balance**

Figure 2 illustrates the water level variations observed at Laguna Honda over the study period. The water level dropped from 1.4 m in February 2021 to 0.6 m in October 2021, then increased stepwise after rain events reaching a maximum level of 1.0 m by the end of April 2022 before decreasing again to 0.6 m in
July 2022. Lake water level variations estimated from hydrological mass balance (Eq. 1), accounting for precipitation on the lake surface, evaporation from the lake surface and basin discharge, agree well with observations over the period of lake water isotope monitoring (model-data deviations are lower than 5 cm) (Fig. 2). This supports that major water sources are integrated, and the lake is disconnected from groundwater aquifers. It further indicates that potential evapotranspiration provides a reliable estimate of
lake evaporation for the studied lake site.

  In the period from January to July 2022, the lake water level is underestimated by about 10 cm, and up to 30 cm in April 2022. This can be attributed to underestimation of *BD*. The TRASERO model predicts *BD* to have occurred only at the beginning of the study period and at end of the rainy season in April 2022. However, water level observations indicated that overland flow or shallow subsurface flow likely
occurred after heavy rainfall events in December 2021 and March 2022 (Fig. 2). Using lower soil WHC in the TRASERO model increases the total amount of *BD*, but none of the modelled scenarios predict *BD* at earlier stages of the rainy season (Fig. A2). This indicates that although the model depicts annual *BD* well, it does not adequately capture the *BD* on a daily scale. Note that basin discharge is negligible during the period of lake water isotope monitoring, so that isotope mass balance calculations are little affected
by missing *BD*.

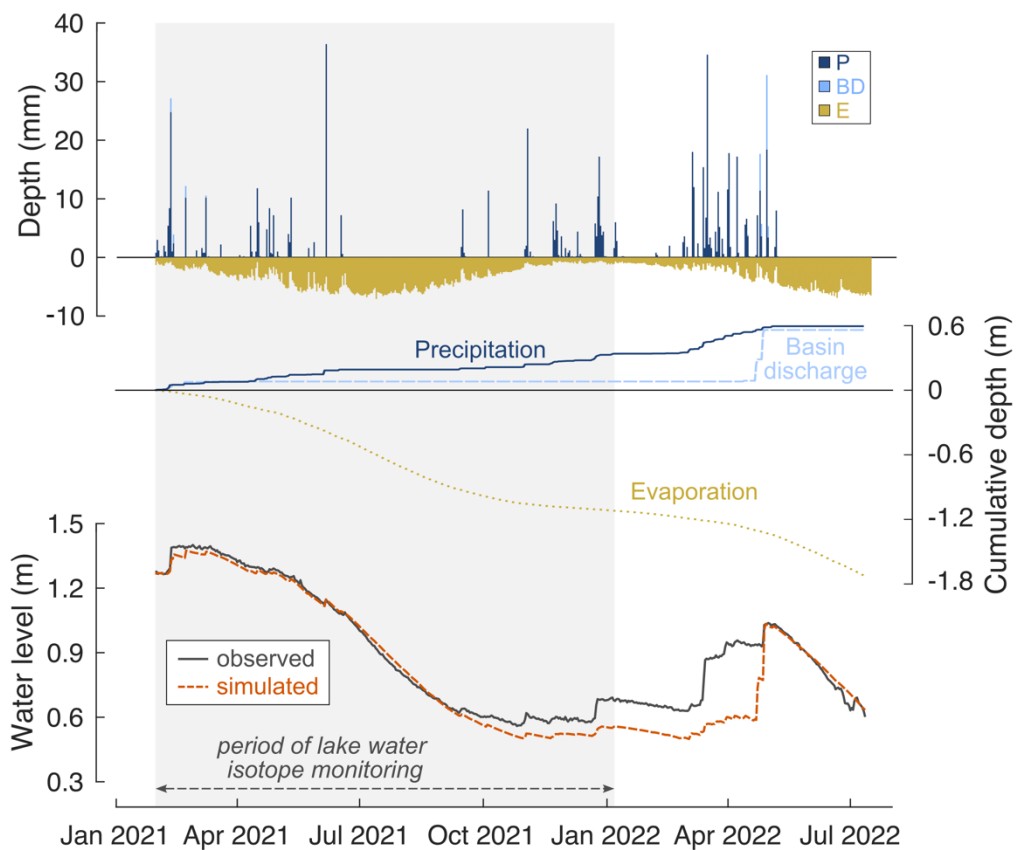

**Figure 2: Simulated and observed water level, depth and estimated cumulative depth of precipitation (P), basin discharge (BD) and evaporation (E) added to and/or removed from the lake over the study period. P and E data are obtained from the meteorological station *Alcaudete* (Junta de Andalucía, 2024). These data were multiplied by the lake surface area to obtain cumulative depth data. BD is simulated for a soil water holding capacity of 190 mm using TRASERO software as described in Section 3.4.1. Lake water volume changes were calculated from hydrological mass balance (Eq. 1) and then converted to simulated water level using the relationship given in Figure 1. See supplementary material for water level simulations using BD simulated for different soil water holding capacities ranging from 130 to 200 mm.**

### 4.3 Water chemistry and salt mass balance

The hydrochemical composition of Laguna Honda is relatively homogenous over the study period and can be classified as Na-Mg-Cl-type (Table A1). Evaporation causes total dissolved solids (TDS) to increase from 23 g L$^{-1}$ in January 2021 to 130 g L$^{-1}$ at the end of the dry period in November 2021. Salinity was spatially homogenous during the rainy season (January to May 2021), but then increased faster at the shallow northern margin, reaching a maximum value of 114 g L$^{-1}$ before the northern part desiccated in July 2021. The concentrations of Na$^+$, K$^+$, Mg$^{2+}$, Cl$^-$, Br$^-$ and SO$_4^{2-}$ correlate with TDS and anticorrelate with the lake water level (Fig. A3), indicating that evaporation is their main driver. The concentration of Ca$^{2+}$ also correlates with TDS until July 2021, but then the trend inverts and the Ca$^{2+}$ concentration starts to decrease despite further increase of TDS. This can be attributed to the precipitation of gypsum and/or anhydrite. The saturation index implies that gypsum precipitation is likely to occur from the end of April,

while saturation with respect to anhydrite is reached in July (Table A3). Further, lake waters are always
saturated with respect to calcite, aragonite and dolomite. In contrast, halite saturation is never reached so
that chloride can be considered as a conservative tracer. The presence of carbonate and sulphate minerals
in lake sediments is evidence of their formation in Laguna Honda (Medina-Ruiz et al., 2024).

Figure 3 illustrates the changes in water level estimated from salt mass balance (Eq. 2) based on the
concentration of chloride. Salt mass balance predicts lake volume (and thus water level) changes similar
to observations, however, underestimates lake volume (water level) from February to September 2021 by
up to 15000 $m^3$ (25 cm). These deviations may be related to chloride concentrations in BD that are higher
than assumed due to dissolution of evaporites in the catchment area during basin discharge.

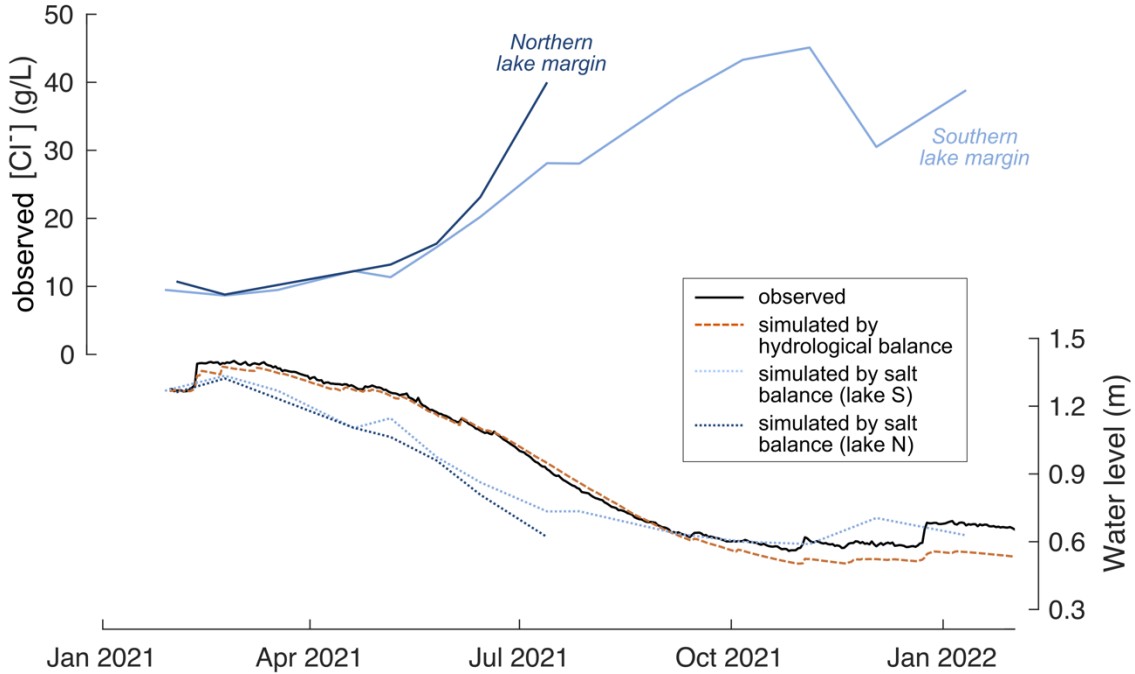

**Figure 3: Time series of chloride concentration, and lake water level over the lake water isotope monitoring period. Upper panel:
Chloride concentration observed at the northern (dark blue) and southern (light blue) lake margin. Lower panel: Observed water
level (black), water level simulated by hydrological balance (red, dashed) using Eq. 1, and water level simulated by salt balance (Eq.
2) using chloride concentrations observed at the northern (dashed, dark blue) and southern (dotted, light blue) lake margin,
respectively. Simulated lake water level was estimated from simulated lake volume using the relationship given in Figure 1.**

**4.4 Isotope data**

Table A4 and A5 presents the results of isotope analysis of precipitation and lake water samples from
Laguna Honda. Precipitation, either directly on the lake surface or indirectly as basin discharge, provides
the major water source for Laguna Honda. Its average amount-weighted isotope composition was -4.3±1.6
‰ for $\delta^{18}O$, -25.8±13.2 ‰ for $\delta^2H$ and 12±13 per meg for [17]O-excess. Values of $\delta^{18}O$ and $\delta^2H$ of
precipitation were lowest in winter (-7.2 ‰ and -47.3 ‰, respectively) and highest in summer (-1.5 ‰
and -5.1 ‰, respectively). Values of [17]O-excess showed the inverse trend, varying between 28 per meg
in winter and -13 per meg in summer. Values of $\delta^{18}O$ and $\delta^2H$ of precipitation are within the range of
global meteoric waters, while its [17]O-excess values are slightly lower than the Global Meteoric Water

Line (Fig. 4), but within the range of recently reported values for precipitation (Aron et al., 2021; Terzer-Wassmuth et al. 2023). The observed $\delta^{18}O$ ($\delta^2H$) of precipitation differed significantly from the values predicted by the Online Isotopes in Precipitation Calculator (Bowen, 2024), particularly in winter (Fig. A4). These deviations may be attributed to large temporal and spatial variability in the frequency and intensity of precipitation as well as in the contribution of moisture sources (Atlantic Ocean, Mediterranean Sea), and moisture recycling. Previous studies suggest that all these parameters influence the isotopic composition of rainfall on the Iberian Peninsula at present (Giménez et al., 2021; Kohfahl et al., 2021) and in the past (Toney et al., 2020).

The lake water showed large temporal, but also spatial isotope variability over the study period (Fig. 4 and 5). Values of $\delta^{18}O$ ($\delta^2H$) of lake water at the southern lake margin increased with decreasing water level from -1.9 ‰ (-26.2 ‰) in January 2021 to 14.9 ‰ (50.7 ‰) at the end of the dry season in September 2021 and subsequently decreased reaching 8.3 ‰ (24.1 ‰) in January 2022. The $\delta^{18}O$ ($\delta^2H$) of lake water at the northern margin was similar to the southern margin from January to May 2021, but then increased at a faster rate, reaching a maximum value of 17.1 ‰ (60.4 ‰) before desiccation in July 2021 (Fig. 5). The $^{17}O$-excess of the lake water generally correlated inversely with $\delta^{18}O$, ranging from -9 per meg to -87 per meg at the southern margin and from -13 per meg to -75 per meg at the northern margin (Fig. 4). The $\delta^{18}O$ and $\delta^2H$ of lake water correlate along a line with a slope of 4.45, characteristic for open-water bodies subject to evaporation (Craig and Gordon, 1965). The line intercepts with the global meteoric water line (GMWL) at -6.6 ‰ and -42.9 ‰ for $\delta^{18}O$ and $\delta^2H$, respectively.

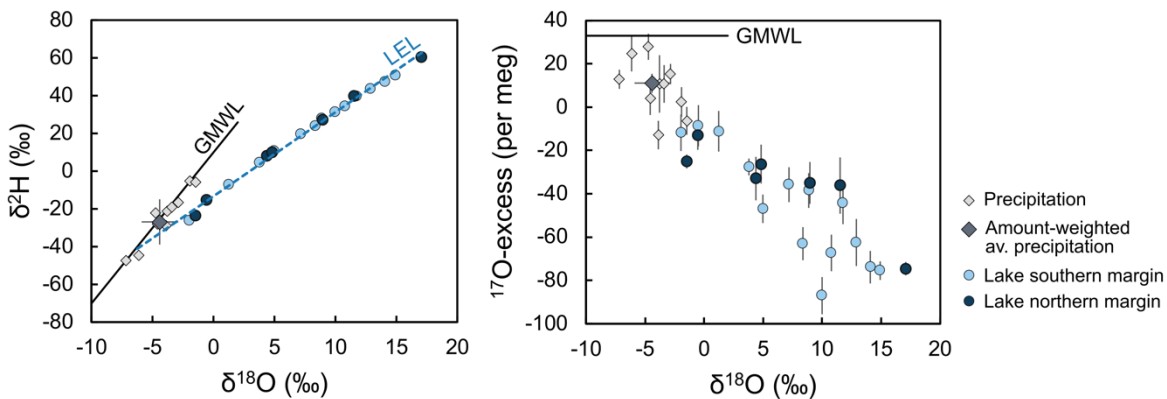

Figure 4: Isotope data of precipitation and Laguna Honda water observed from January 2021 to January 2022. The black solid line represents the Global Meteoric Water Line (GMWL), while the dashed blue line is the regression line through lake water isotope data that represents the local evaporation line (LEL).

**4.5 Isotope mass balance**

The isotope mass balance model reflects the general trends of the lake water isotope composition observed over the study period, but overestimates $\delta^{18}O$ ($\delta^2H$) of lake water during the dry season up to 5.0 ‰ (21.3 ‰) and underestimates the observed $\delta^{18}O$ ($\delta^2H$) of lake water during the rainy season up to 3.2 ‰ (14.3 ‰) in May 2021 and up to 7.9 ‰ (36.2 ‰) in January 2022 (Fig. 5a and b). The simulated $^{17}O$-excess of lake water roughly agrees with observations from January to July 2021 but then starts to underestimate the observed values reaching a maximum deviation of 94 per meg at the end of the dry season in October 2021 (Fig 5c).

Sensitivity experiments showed that misestimation of the initial lake water level has relatively little influence on the simulated lake water isotope composition, compared to the large offset observed between the isotope model and the data (Fig. A5 left panel). Using an initial water level of ±0.1 m of the observed value resulted in $\delta^{18}O$, $\delta^{2}H$ and $^{17}O$-excess that deviated in average only 0.7 ‰, 2.5 ‰ and 4 per meg from the original simulation and never exceeded deviations of 1.8 ‰, 7.5 ‰ and 23 per meg, respectively.

In general, using a higher initial water level results in less seasonal isotope variability of lake water, i.e. reduces the model-data offset. In contrast, using a lower initial lake water level, seasonal isotope variations of lake water are more pronounced, i.e. the model-data offset is increased. The impact of misestimation of the lake surface area-to-volume ratio on the simulated lake water isotope composition is on similar magnitude as that of the initial water level (Fig. A5, right panel). Varying the lake surface

area-to-volume ratio by ±0.1 around the value estimate from the lake bathymetric model (Fig. 1) leads to variations of simulated $\delta^{18}O$, $\delta^{2}H$ and $^{17}O$-excess of lake water of only 0.5 ‰, 2.2 ‰ and 5 per meg in average, that reach 1.1 ‰, 4.6 ‰ and 13 per meg at maximum. Lowering the lake surface area-to-volume ratio reduces the amplitude of simulated seasonal isotope variations of lake water, while larger seasonal isotope variations of lake water are simulated when a higher lake surface area-to-volume ratio is used.

Both the sensitivity of the lake water isotope composition to the initial water level and the lake surface area-to volume ratio can be attributed to changes in the ratio of the evaporation flux to total lake volume and explained considering mass balance. Lowering the initial water level implies a lower total lake volume and, for Laguna Honda, a higher lake surface area-to-volume ratio, which, in turn, increases the proportion of evaporation on the total lake volume, and thus the amplitude of seasonal isotope variations

of lake water.

  In contrast to uncertainty in lake bathymetry, changes in the isotope composition of the evaporation flux, driven by relative humidity, the turbulence coefficient, and the isotope composition of atmospheric water vapor can strongly influence the isotope composition of lake water (Fig. A6). The $\delta^{18}O$ ($\delta^{2}H$) of water vapor in equilibrium with precipitation varies mostly between -14 and -12 ‰ (-110 to -80 ‰) and reaches

lowest values in December 2021 and January 2022 (-17 and -18‰). The $^{17}O$-excess of water vapor in equilibrium with precipitation ranges between -23 per meg and 14 per meg, significantly lower than the average of global meteoric waters. Assuming that the isotope difference between precipitation and atmospheric water vapor is 25 % lower (higher) than at isotope equilibrium, increases (decreases) values of $\delta^{18}O$, $\delta^{2}H$ and $^{17}O$-excess of atmospheric water vapor by 2.5 ‰, 18.1 ‰ and 6 per meg, respectively.

Varying the isotope composition of atmospheric water vapor between these limits causes variations in the simulated $\delta^{18}O$, $\delta^{2}H$ and $^{17}O$-excess of lake water up to 2.4 ‰, 19.8 ‰ and 12 per meg, respectively (Fig. A6, left panel). An increase in the isotope difference between precipitation and atmospheric water vapor causes simulated $\delta^{18}O$ and $\delta^{2}H$ of lake water to decrease, while simulated $^{17}O$-excess of lake water is increased. Setting the $^{17}O$-excess of atmospheric water vapor to a constant value of 22 per meg, increases

the simulated $^{17}O$-excess of lake water by up to 38 per meg, with highest impact in summer months (Fig. A6, left panel, yellow line in c). The turbulence coefficient has only little influence on the $\delta^{2}H$ but can strongly affect the $\delta^{18}O$ and $^{17}O$-excess of the simulated lake water. Using a turbulence coefficient of 0.6 instead of 0.5 increases the $\delta^{18}O$ of the simulated lake water by up to 2.8 ‰, while the $\delta^{2}H$ of the simulated lake water increases by only 2.6 ‰ at its maximum (Fig A6, central panel). Inversely, the $^{17}O$-excess of

the simulated lake water decreases by up to 42 per meg. The relative humidity has the strongest influence on the lake water isotope composition. A ±5 % deviation of the local relative humidity at the lake site

from that observed at the meteorological station causes the simulated $\delta^{18}O$, $\delta^2H$ and $^{17}O$-excess of lake water to deviate up to 10.5 ‰, 48.6 ‰ and 92 per meg from the original simulation, respectively (Fig. A6, right panel).

No model-data agreement is found for a constant set of relative humidity, the turbulence coefficient and the isotope composition of atmospheric water vapor. Instead, seasonal variations in all three parameters need to be taken into account. Considering the results of the sensitivity experiments, better model-data agreement in the rainy season can achieved when using a relative humidity that is lower than the observed value and a slightly higher turbulence coefficient (Fig. 5a-b, yellow curve). In contrast, during the dry
season, using a higher relative humidity value and lower turbulence coefficient is necessary to reduce the offset between the modelled and observed isotope composition of lake water (Fig. 5a-b, yellow curve). The simulated $^{17}O$-excess of lake water coincides only with observations when using a higher $^{17}O$-excess of atmospheric water vapor (33 per meg) (Fig. 5c, yellow curve).

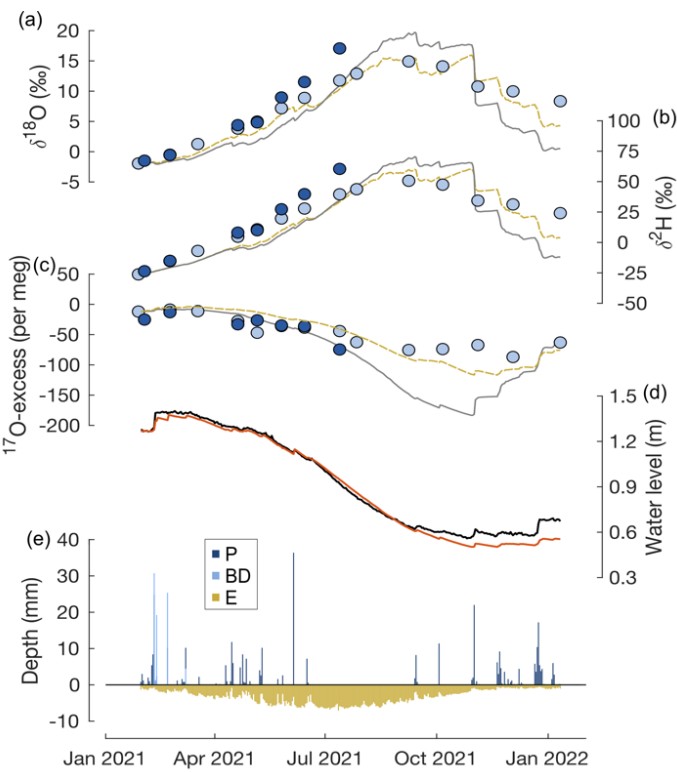

**Figure 5: Time series of (a) $\delta^{18}O$, (b) $\delta^2H$ and (c) $^{17}O$-excess of lake water sampled at the southern (light blue) and northern (dark blue) margins of Laguna Honda, (d) observed (solid, black) and simulated (solid, red) water levels, (e) simulated inflow from precipitation (P) and basin discharge (BD) and outflow by evaporation (E). In panels (a)-(c), the solid black line represents the simulated isotope mass balance calculated using observed atmospheric relative humidity, a turbulence coefficient of 0.5 and assuming the isotope composition of atmospheric water vapor to be in isotope equilibrium with precipitation. The dashed yellow line**
**shows an alternative simulation that reduces the deviation between observed and simulated data. For this simulation, the dataset was divided in a rainy (January to May 2021 and October 2021 to January 2022) and a dry season (June to September 2021). Values of $V_{eq} = 1$, n = 0.6, and observed RH – 5% were used for the rainy season and $V_{eq} = 1$, n = 0.4 and observed RH +5 for the dry season. Analytical errors are smaller than symbol size.**

**5 Discussion**

 **5.1 Changes in relative humidity control seasonal isotope variations of lake water in non-recharged lake systems**

The isotope composition of lake systems that discharge groundwater aquifers, i.e. continuously receive inflow from groundwater, is controlled by the ratio between evaporation and inflow. Many studies have taken advantage of this to estimate the evaporation-to-inflow ratio from lake water isotope compositions (e.g., Gibson et al., 2016b; Jasechko et al., 2014; Zanazzi et al., 2020; Zuber, 1983). However, small lakes
systems in semiarid environments that undergo substantial water level fluctuations on seasonal and interannual scale, are often disconnected from groundwater aquifers. In such systems, such as Laguna Honda, evaporation is the major driver of the isotope composition of lake water, while precipitation and basin discharge cause only short transitional mixing events, in which relatively unevaporated meteoric water is admixed to the evaporated lake water. Even if the amount of precipitation per event can exceed
the evaporation rate per day by more than one magnitude, its volume relative to the total lake volume is usually low, so that its impact on the isotope composition of the lake water is often negligible. In the case of Laguna Honda, the impact of precipitation on the lake water isotope composition was usually less than 2 % and always lower than 7 %. The impact of precipitation can, however, become significant at low lake water levels (< 0.5 m).
Evaporation occurs continuously and thereby drives the lake water isotope composition towards an isotope steady state value that is principally controlled by the atmospheric relative humidity, the isotope composition of atmospheric water vapor and atmospheric turbulence (Craig and Gordon, 1965). How fast the lake water approaches this isotope steady state depends on the proportion of water loss by evaporation to the lake volume. Isotope variations of lake systems in semi-arid environments are thus highest in
summer when evaporation rates are highest, and water level is lowest (i.e. the surface area-to-volume ratio is higher). However, the isotope steady state value is only approached slowly in our study case. On a daily scale, the water loss due to evaporation is typically less than 1 % of the lake volume and never exceeds 2.5 %. As the isotope steady state value is subject to continuous changes due to changes in environmental conditions, especially relative humidity, it is rarely reached – often only by chance, when
changes in relative humidity cause the isotope steady state value to approach that of the lake water. The isotope steady state value is in isotope equilibrium with atmospheric water vapor at atmospheric moisture saturation ($h = 100$ %) and moves to higher $\delta^{18}O$ ($\delta^2H$) and lower $^{17}O$-excess values with decreasing relative humidity. Therefore, the $\delta^{18}O$ ($\delta^2H$) of Laguna Honda water increased continuously from winter to summer, while $^{17}O$-excess decreased. With the beginning of the subsequent rainy season, the relative
humidity increases and consequently the $\delta^{18}O$ ($\delta^2H$) of Laguna Honda water decreased again.
The isotope mass balance model is highly sensitive to the parameters controlling the isotope composition of the evaporation flux – relative humidity, the isotope composition of atmospheric water vapor, and the turbulence coefficient. An accurate estimation of these parameters is however challenging. Adapting the isotope model to the observed isotope data can reveal information on these parameters. For example,
during the rainy season, the model requires a relative humidity that is lower than the observed value to achieve a better agreement between the modelled and the observed isotope composition of lake water. The lower relative humidity is likely related to the diurnal cycle of evaporation. In general, evaporation is higher during daytime, so that the effective relative humidity during evaporation (weighted by the

evaporation rate) is lower than its daily average value. Observations show that the difference between nighttime and daytime relative humidity is strongest in winter (up to 7 %), when relative humidity is high. In summer, relative humidity between day and night vary only by about 3 %. Further, relative humidity may be overestimated during the rainy season as precipitation leads to higher atmospheric relative humidity. Indeed, relative humidity on rainy days averages to 82 %, while the average on non-rainy days is 59 %. Evaporation is more pronounced on non-rainy days and thus a lower relative humidity may drive the overall isotope composition of the evaporation flux. In contrast, in summer, the model requires a relative humidity higher than the observed value to achieve a better agreement between the modelled and the observed isotope composition of lake water. This counterintuitive observation may result from strong evaporation during summer, which increases atmospheric relative humidity. The upward evaporation flux contributes to moisture buildup and perturbs the atmospheric boundary layer. This perturbation can be accounted for in the model by weighting the turbulence coefficient with a transport parameter, $\theta$ (Gat et al., 1994; Gibson et al., 2016b). For the Great lakes region, Gat et al. (1994) found $\theta$ to be 0.88. However, values closer to 1 are expected for smaller lakes (Gibson et al., 2016b). Alternatively, the lower turbulence coefficient in summer may result from stronger atmospheric turbulence caused by a higher frequency of windy days.

Lake water salinity affects isotope activities and increases fluid viscosity and may thereby influence isotope fractionation during evaporation. In the model simulation, this influence can be accounted for by correcting equilibrium fractionation factors for the classical salt effect (Horita, 1989, 2005; Horita et al., 1993) and by incorporating effective rather than actual relative humidity. Simulation experiments demonstrate that the classic salt effect and the viscosity-induced changes in effective relative humidity roughly compensate each other. The salinity is, therefore, of minor importance for the isotopic composition of the lake water. Simulation tests further showed that the precipitation of hydrated minerals such as gypsum has a negligible influence on lake water's isotope composition. Unlikely large amounts of several hundred of kilograms of gypsum would need be precipitated from the lake water to explain the differences between the modelled and observed lake water isotope composition in summer.

**5.2 The value of triple oxygen isotopes**

Our results show how the triple oxygen isotope composition of a semi-permanent lake, which is only occasionally recharged by precipitation and basin discharge, evolves in a semiarid environment. Figure 6a illustrates the evolution of $^{17}$O-excess vs. $\delta'^{18}$O of the lake water over the study period as well as the variability of the theoretical isotope steady state value in dependence of the relative humidity. At the beginning of the record, the isotope composition of the lake water shows lowest $\delta'^{18}$O and highest $^{17}$O-excess of the overall record, driven by a steady state value that is at the upper end of the trend due to prevailing high relative humidity (Fig. 6a). Towards the dry season, relative humidity decreases, and the theoretical isotope steady state value evolves exponentially to higher $\delta'^{18}$O and lower $^{17}$O-excess values. The $\delta'^{18}$O and $^{17}$O-excess of the lake water do not evolve along the linear trend of the isotope steady state value but form a convex curvature as predicted from isotope theory (Fig. 6a). The lake water reaches highest $\delta'^{18}$O and lowest $^{17}$O-excess at the end of the dry season, when relative humidity is lowest. With the beginning of the rainy season, the relative humidity increases, so that, for the first time in our record, the theoretical isotope steady state value has lower $\delta'^{18}$O and higher $^{17}$O-excess than the lake water.

During evaporation, the isotope composition of the lake water evolves now along a concave curvature
towards this isotope steady value (Fig. 6a). A similar looping trend was observed in plant waters on a
daily scale and attributed to the day-night variability of the isotope steady state value (Voigt et al., 2023).
The lake volume is several magnitudes higher than that of plant leaves, so that day-night changes in
relative humidity do not affect the lake water isotope composition. However, lake water residence time is
low enough to reflect seasonal climate variations.

Admixture of precipitation to the evaporated lake water at the beginning of the subsequent rainy season
can cause trends similar to those observed in our study (Voigt et al., 2021). However, the contribution of
precipitation and basin discharge to the total lake volume is always lower than 10 % in our study case,
influencing $\delta^{18}O$, $\delta^2H$ and $^{17}O$-excess of lake water by less than 1.0 ‰, 1.9 ‰ and 8 per meg, respectively.
The admixture of precipitation or basin discharge can therefore not fully explain the observed shifts in
the isotopic composition of the lake water. Moreover, such mixing effects are of transient nature. If
sampling is not carried out immediately after the rain event, the mixing signature is overprinted by
subsequent evaporation. In a simulation experiment in which precipitation and basin discharge were
neglected and only evaporation was considered, the isotopic composition of lake water showed a looping
trend similar to the observed one. This supports that shifts in the theoretical isotope steady state value
drive the seasonal isotope variations of lake water. The triple oxygen isotope system allows the
identification of these non-steady state conditions caused by seasonal changes in environmental
conditions, especially relative humidity. This is not always evident using $\delta^{18}O$ and $\delta^2H$ alone, due to the
linearity of evaporation trends in this isotope space (Fig. 6b).

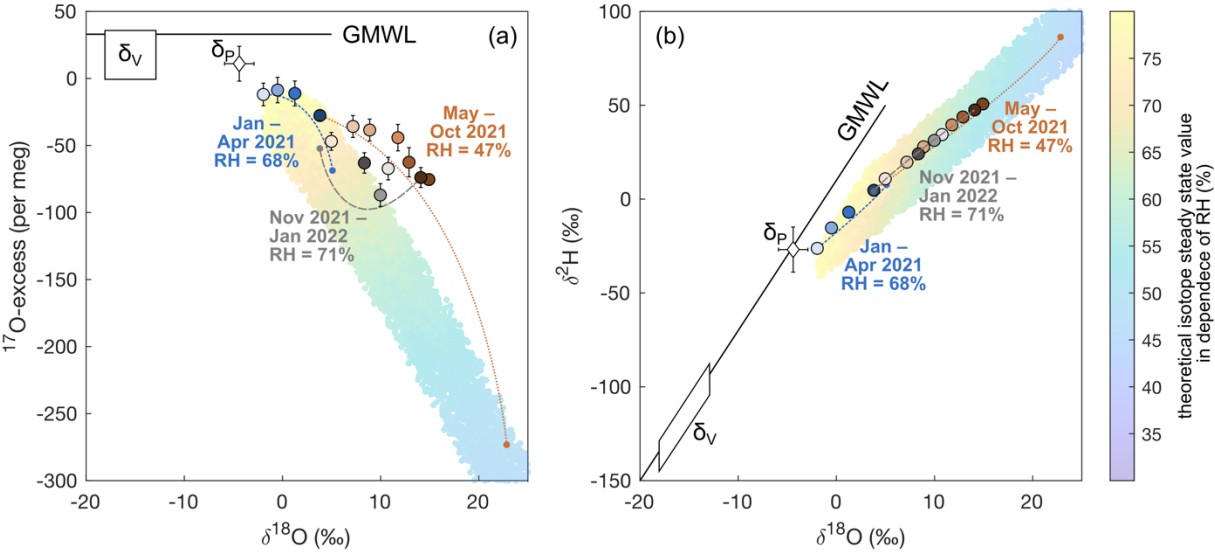

**Figure 6: Comparison of the evolution of the isotope composition of lake water to theoretical evaporation trajectories simulated for
average conditions during the rainy and dry periods in diagrams of (a) $^{17}O$-excess over $\delta'^{18}O$ and (b) $\delta^2H$ over $\delta^{18}O$. Filled circles
show the isotope composition of lake water with colours representing samples taken during rainy period (January to April 2021,
blueish) with an average relative humidity (h) of 68 % and temperature (T) of 12 ºC, the dry period (May to October 2021, brownish)
with an average h of 47 % and T of 23 ºC, and at the beginning of the subsequent rainy season (November 2021 to January 2022,
greyish) with an average h of 71 % and T = 10 ºC. The colour gradient indicates the evolution of the lake water isotope composition
within each time block. The theoretical evaporation trajectories (dashed lines) are determined based on the C-G model for simple**

evaporation (Craig and Gordon, 1965; Surma et al., 2015) considering the seasonal variations in h and T given above, and using $\delta^{18}O_V$ of atmospheric water vapor of -17 ‰ during the rainy period and -14 ‰ during the dry period and global meteoric values for
$^{17}O$-excess$_V$ and d-excess$_V$ of atmospheric water vapor (33 per meg and 10 ‰, respectively). The evaporation trajectories illustrate the theoretical evolution of the isotope composition of water undergoing evaporation towards the isotope steady state value (small dots at the end of the curves) in dependence of the residual fraction. This isotope steady state value is never reached as only a small fraction of the total lake volume evaporates in each time step. The colour band in the background shows the isotope composition of the theoretical isotope steady state value (Eq. 6) in dependence of RH, simulated using an annual average temperature of 17ºC, a
turbulence coefficient of 0.5 and varying $\delta^{18}O$, d-excess and $^{17}O$-excess of atmospheric water vapor between -18 ‰ and -13 ‰, 0 ‰ and 15 ‰, and 0 per meg and 35 per meg, respectively. The global meteoric water line (GMWL) and the amount-weighted average isotope composition of precipitation ($\delta_P$) are shown as reference.

### 5.3 Hydrological functioning of Laguna Honda and future developments

Agreement of simulated and observed water level changes imply that precipitation, basin discharge and
evaporation drive the water balance of Laguna Honda. Similar conclusions have been drawn from annual hydrological mass balance calculations of Laguna Honda by Moral et al. (2008). The drainage basin is fundamentally impermeable, so that groundwater discharge is unlikely to occur. However, carbonate formations and detritic Neogene and Quaternary sediments in the drainage basin could constitute aquifers, from which an additional supply of strongly mineralized groundwater could be obtained (Medina-Ruiz et
al., 2024; Moral et al., 2008). Nevertheless, a substantial discharge (recharge) of the underlying groundwater aquifer would result in a significant imbalance, whereby the simulated water level would be significantly lower (higher) than the observed one. This has been observed for some dune ponds in the Doñana National Park, SE Spain (Fernández-Ayuso et al. 2018). A seasonal inflow from basin discharge that may be caused by irrigation of the surrounding olive fields cannot be ruled out, but if so, the amount
is small, so that the water balance is not significantly affected. Throughflow of groundwater from a carbonate aquifer would not affect the hydrological balance, but if so, the groundwater flow associated with these permeable blocks is low, as they are generally small in size (Moral et al., 2008). In addition, the large seasonal variations in salinity of Laguna Honda (23-130 g L$^{-1}$) are not consistent with throughflow conditions that would cause only little salinity fluctuations.
The spatial differences in the isotope composition of lake water that reach up to 5.3 ‰, 20.9 ‰ and 30 per meg, can be attributed to the asymmetric morphometry of the lake basin (Fig. 1b). The northern margin of the lake is characterized by gentle slopes and markedly less deep than the southern margin. Therefore, when the water level drops, the influence of evaporation on the isotopic and chemical composition of water in the shallow northern part of the lake is stronger than in the deeper southern part.
The limited mixing in summer reinforces this spatial heterogeneity. In contrast, in winter, the lake is deeper and thus the wind fetch higher, so that the lake is well-mixed. Vertical stratification of the lake's water column in summer cannot be completely ruled out but is unlikely given the shallow depth of the lake.
Spatial variations in the isotope composition of lake water should be considered when interpreting isotope
data obtained from paleo-archives from lake sediments. Recently, gradients in the isotope composition of gypsum hydration water from the lake margin to the centre have been linked to increasing lake water evaporation due to decreasing water level (Cañada-Pasadas et al., 2024). Sampling lake sediments along horizontal transects can thus provide information at different evaporation stages.
During the study period, the lake's surface evaporation exceeded the combined water inputs from
precipitation and basin discharge by more than twofold, resulting in a significant water level decrease of

approximately 0.6 m. Analysis of Copernicus Sentinel-2 satellite imagery from 2016 to today indicates that the water level of Laguna Honda has declined by about 1 m over the past decade. By the end of July 2024, the water level has already dropped below 0.3 m, suggesting the lake may potentially desiccate entirely during the dry season. The negative water balance, exacerbated by ongoing anthropogenic climate
change, is expected to cause more frequent and prolonged desiccation events in Laguna Honda in the future. This trend has critical implications for the flora and fauna dependent on this wetland for survival (Jiménez-Melero et al., 2012, 2023; Moral et al., 2008).

## 6 Conclusion

We assessed the hydrogeochemical functioning of Laguna Honda, a small, semi-permanent lake situated
within the Andalusian lowlands using hydrological, isotope and chemical tools. The results demonstrated that the lake's hydrological balance is mainly driven by evaporation, precipitation and basin discharge. Over the 18-month study period, water loss by evaporation from the lake surface was found to be more than twice that of water input by precipitation directly on the lake surface and indirectly from basin discharge. This resulted in a reduction of the lake water level by 0.8 m. This emphasizes the impact in
view of climate change that is particularly pronounced in the Mediterranean region and highly threatens many Andalusian lakes to desiccate more frequently.

The results further revealed that the isotope composition of lake water is mainly controlled by evaporation, while the impact of precipitation and basin discharge are only of transitional nature. For the first time, a non-steady state isotope mass balance model has been applied for triple oxygen isotopes. Our
results demonstrate that the evolution of $^{17}$O-excess over $\delta'^{18}$O in lake water forms a loop that is driven by seasonal changes in the theoretical isotope steady state value. This, in turn, is driven by environmental conditions, in particular relative humidity. The significant seasonal isotope variability underscores the importance of sampling timing in the assessment of lake water balances. One-time or short-term lake water sampling can lead to inaccuracies, especially if conducted at the beginning of the rainy season when
deviations from steady-state conditions are most pronounced.

Sensitivity analyses showed that the isotope composition of atmospheric water vapor is a key parameter determining isotope fractionation of lake water during evaporation. In the absence of direct measurements, it is often estimated from precipitation data assuming isotope equilibrium. However, in semi-arid regions, the seasonality of precipitation challenges accurate estimation of the isotope
composition of atmospheric water vapor. Establishing long-term records of atmospheric water vapor isotopes, particularly in regions of low or highly seasonal rainfall, would help to better constrain this variable, enhancing model simulations and reducing uncertainty in lake isotope mass balance calculations. Our results have significant implications for interpreting isotope data retrieved from lake sediment archives in paleoclimate studies. Lake evaporites, such as gypsum, capture the isotopic composition of
the lake water at the time of their formation. Recently, Cañada-Pasadas et al. (2024) demonstrated that surface gypsum samples of an Andalusian wetland reflect lake water isotope conditions during spring and early summer, rather than the annual average. Determining the timing of gypsum formation is thus essential to ensure accurate paleo-interpretations. Additionally, spatial variations in the isotope

composition of lake water must be considered. Lake sediment sampling along horizontal transects can reveal information on paleo-environmental conditions at different evaporation stages.

**Appendices**

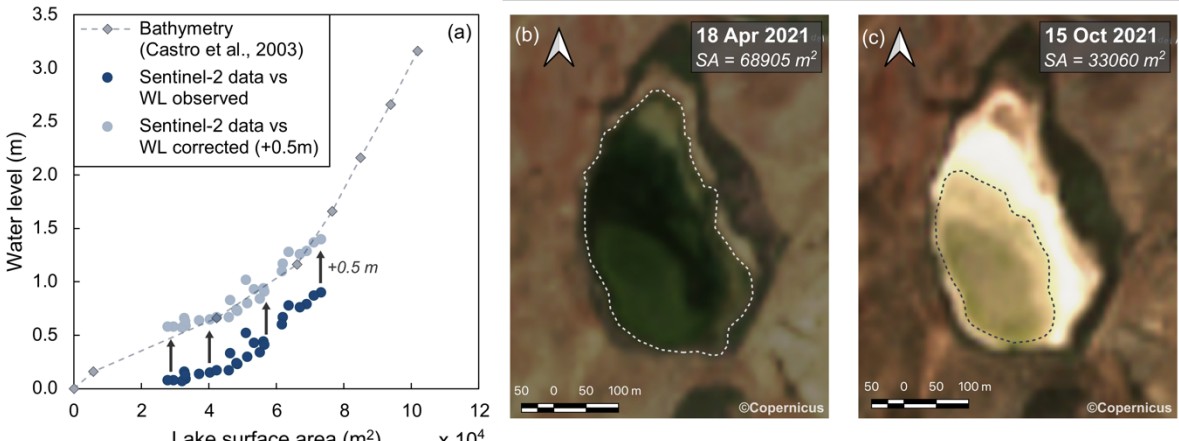

Figure A1: (a) Comparison of lake surface area estimated from Copernicus Sentinel-2 satellite imagery to the water level-to-lake surface area relationship obtained from lake bathymetry (Fig. 1, Castro et al. 2003). Estimated lake surface area coincides to water levels (WL) that are about 0.5m higher than observed lake water levels. (b) and (c) show Copernicus Sentinel-2 satellite imagery exemplary for high and low water level, respectively. The dashed lines illustrate the outline of the flooded surface area (SA), contoured using the Free and Open Source software QGIS.

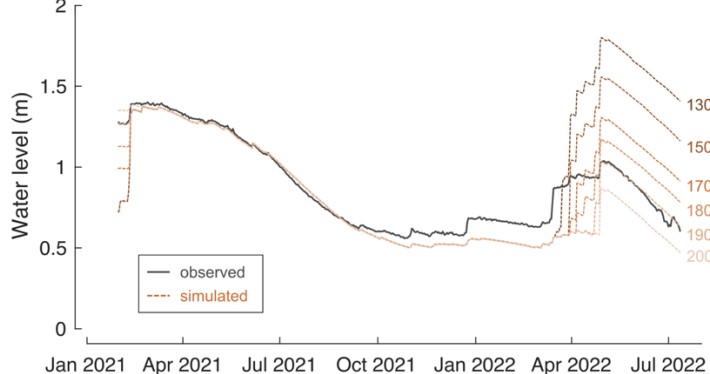

Figure A2: Observed and simulated water level calculated using basin discharge simulated for soil water capacities ranging from 130 to 200 mm (small numbers).

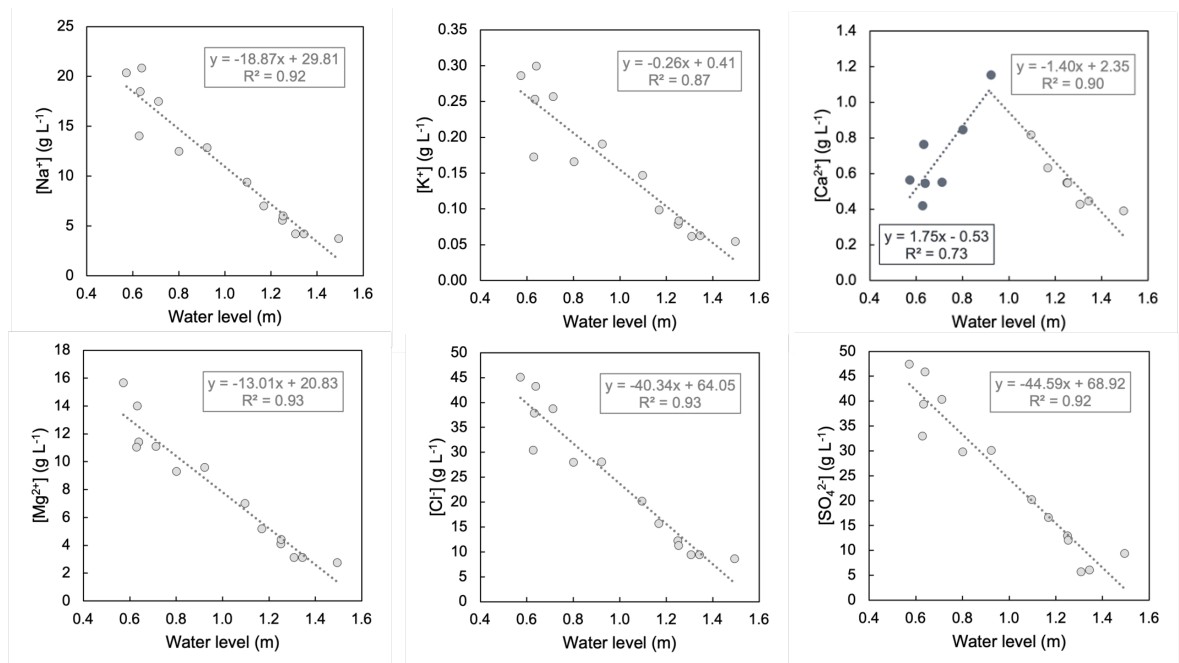

**Figure A3: Relationship of the concentration of major ions (Na⁺, K⁺, Ca²⁺, Mg²⁺, Cl⁻ and SO₄²⁻) to the water level during the study period. Only lake water samples from the southern margin are considered. All ion concentration data gathered in this study are given in Table A3.**

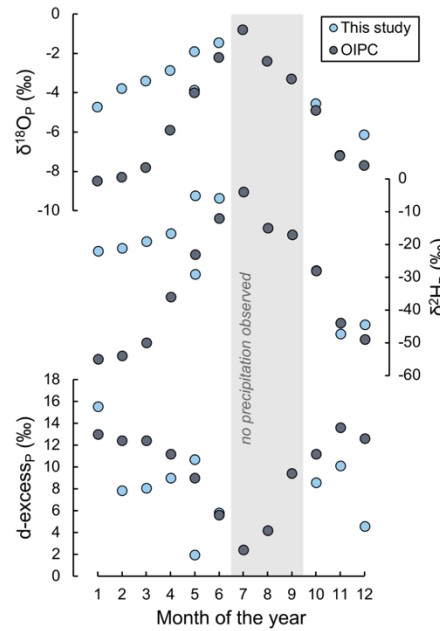

**Figure A4: Comparison of amount-weighted monthly average δ¹⁸O, δ²H and d-excess of precipitation observed at Laguna Honda over the study period and values predicted by the Online Isotopes in Precipitation Calculator for the study site (OIPC, 2024). No precipitation has been observed in July, August and September 2021.**

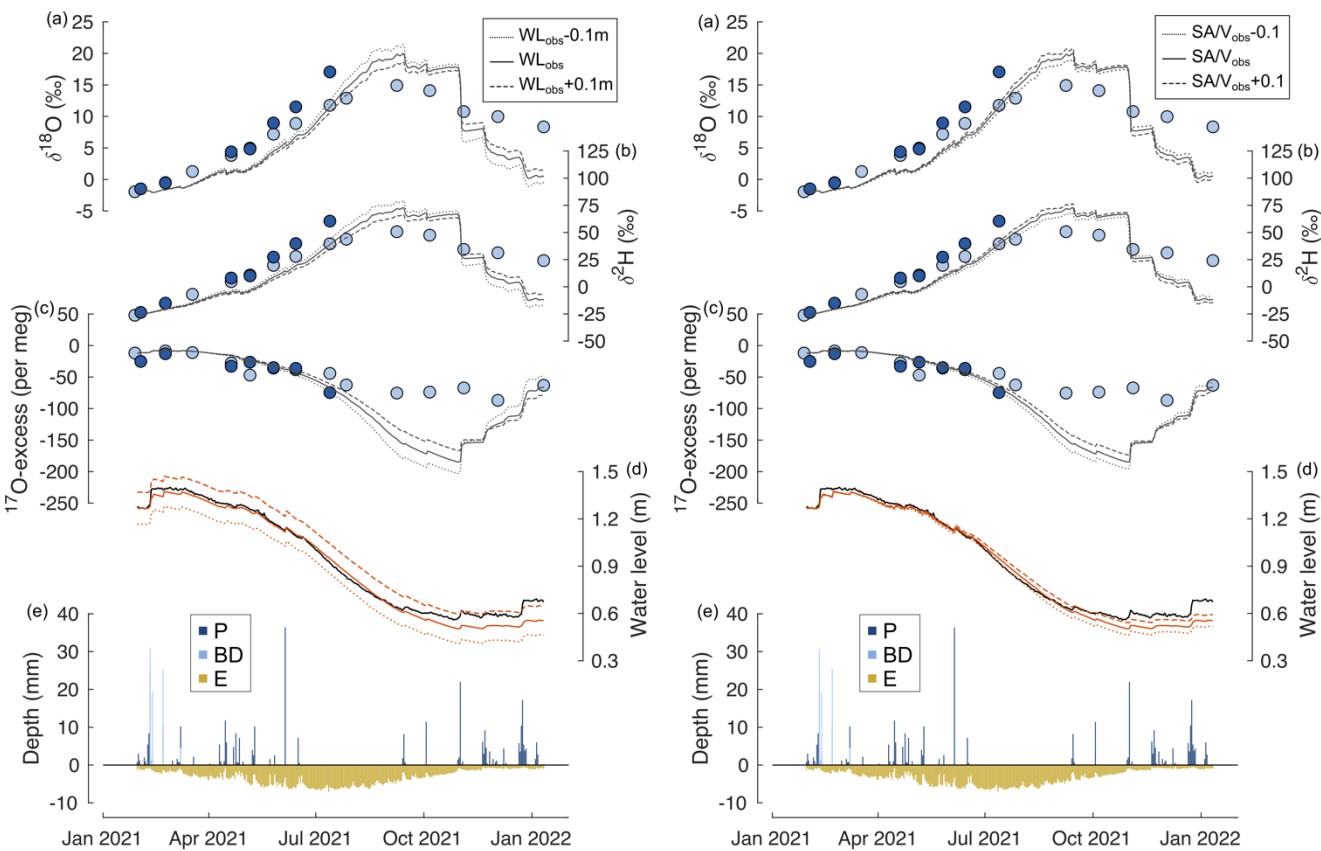

Figure A5: Illustration of the sensitivity of the isotope mass balance simulation to the initial lake water level. Time series of (a)-(c) $\delta^{18}O$, $\delta^2H$ and $^{17}O$-excess of lake water sampled at the southern (light blue) and northern (dark blue) margin of Laguna Honda, (d) observed (solid, black) and simulated (dashed, red) water level, (e) simulated inflow from precipitation (P) and basin discharge (BD) and outflow by evaporation (E). The solid black line shows the simulated isotope composition of lake water using observed atmospheric relative humidity, assuming the isotope composition of atmospheric water vapor to be in isotope equilibrium with precipitation and a turbulence coefficient of 0.5. Left panel (a)-(c): The dashed (dotted) black line shows the simulated isotope composition of lake water using similar input parameters as for the solid line but an initial water level that is 0.1m higher (lower) than the observed water level. Right panel (a)-(c): The dashed (dotted) black line shows the simulated isotope composition of lake water using similar input parameters as for the solid line but a surface-area-to-volume ratio that is 0.1 higher (lower) than the observed one.

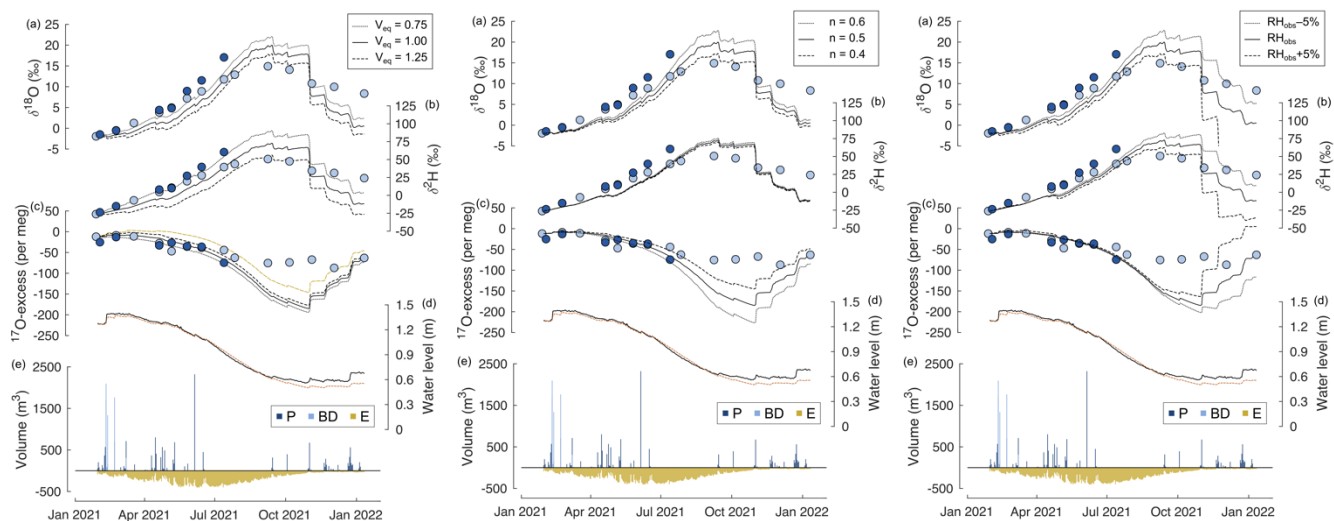


**Figure A6: Illustration of the sensitivity of the isotope mass balance simulation to the isotope composition of the evaporation flux.**
Time series of (a)-(c) $\delta^{18}O$, $\delta^2H$ and $^{17}O$-excess of lake water sampled at the southern (light blue) and northern (dark blue) margin of Laguna Honda, (d) observed (solid, black) and simulated (dashed, red) water level, (e) simulated inflow from precipitation (P) and basin discharge (BD) and outflow by evaporation (E). The solid black line shows the simulated isotope composition of lake water

using observed atmospheric relative humidity, assuming the isotope composition of atmospheric water vapor to be in isotope equilibrium with precipitation and a turbulence coefficient of 0.5. Left panel (a)-(c): The dashed (dotted) line shows the simulated isotope composition of lake water using similar input parameters as for the solid line but assuming the isotope composition of atmospheric water vapor to be 25% higher (lower) than expected from isotope equilibrium. The yellow dashed line in (c) shows the simulated $^{17}O$-excess of lake water using similar input parameters as for the solid line but assuming $^{17}O$-excess of atmospheric water

vapor to be constant (22 per meg). Central panel (a)-(c): The dashed (dotted) line shows the simulated isotope composition of lake water using similar input parameters as for the solid line but assuming higher (lower) atmospheric turbulence, accounted for by varying turbulence coefficient. Right panel (a)-(c): The dashed (dotted) line shows the simulated isotope composition of lake water using similar input parameters as for the solid line but using a 5% higher (lower) relative humidity than observed at the nearby meteorological station.

**Data availability**

The authors confirm that the data supporting the findings of this study are available within the article [and/or] its appendices.

**Author Contribution**

Data collection: AM, RJE, JJM, AISV; analysis and interpretation of the results: CV, FG, MRR, LM;
draft manuscript preparation: CV, FG. All authors reviewed the results and approved the final version of the manuscript.

**Competing Interests**

The authors declare that they have no conflict of interest.

## Acknowledgements

This study was supported by project PID2021-123980OA-I00 (GYPCLIMATE), funded by the Ministerio de Ciencia e Innovación of Spain, the Agencia Estatal de Investigación and the Fondo Europeo de Desarrollo Regional FEDER. CV was supported by the European Commission (Marie Curie postdoctoral fellowship, grant no. 101063961). LM is funded by the FPU21/06924 grant of the Spanish *Ministerio de Educación y Formación Profesional*. FG acknowledges the Ramón y Cajal fellowship,
RYC2020-029811-I and the grant PPIT-UAL, Junta de Andalucía-FEDER 2022-2026 (RyC-PPI2021-01).

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

**Table A1:** Hydrochemical composition of sampled lake waters sampled at Laguna Honda. TDS = total dissolved solids, calculated as the sum of measured cation and anion concentrations; bdl = below detection limit.

| Sampling Date | Water level (m) | Na$^+$ (g L$^{-1}$) | K$^+$ (g L$^{-1}$) | NH$_4^+$ (g L$^{-1}$) | Mg$^{2+}$ (g L$^{-1}$) | Ca$^{2+}$ (g L$^{-1}$) | Cl$^-$ (g L$^{-1}$) | F$^-$ (g L$^{-1}$) | Br$^-$ (g L$^{-1}$) | NO$_3^-$ (g L$^{-1}$) | SO$_4^{2-}$ (g L$^{-1}$) | TDS (g L$^{-1}$) |
|---|---|---|---|---|---|---|---|---|---|---|---|---|
| *Southern lake margin* | | | | | | | | | | | | |
| 28/1/21 | 1.31 | 4.2 | 0.06 | 0.0006 | 3.1 | 0.43 | 9.5 | 0.002 | 0.021 | 0.047 | 5.7 | 23.0 |
| 23/2/21 | 1.50 | 3.7 | 0.05 | 0.0004 | 2.7 | 0.39 | 8.7 | 0.001 | 0.028 | 0.040 | 9.3 | 25.0 |
| 18/3/21 | 1.34 | 4.2 | 0.06 | 0.0021 | 3.1 | 0.45 | 9.5 | 0.002 | 0.024 | 0.045 | 6.0 | 23.4 |
| 20/4/21 | 1.25 | 5.5 | 0.08 | 0.0006 | 4.1 | 0.55 | 12.3 | 0.003 | 0.039 | 0.059 | 12.9 | 35.5 |
| 6/5/21 | 1.25 | 6.0 | 0.08 | bdl | 4.4 | 0.55 | 11.3 | 0.003 | 0.039 | 0.067 | 12.0 | 34.5 |
| 26/5/21 | 1.17 | 7.0 | 0.10 | 0.0009 | 5.2 | 0.63 | 15.8 | 0.003 | 0.053 | 0.078 | 16.6 | 45.4 |
| 14/6/21 | 1.10 | 9.4 | 0.15 | 0.0013 | 7.0 | 0.82 | 20.2 | 0.009 | 0.067 | 0.107 | 20.2 | 57.9 |
| 13/7/21 | 0.92 | 12.8 | 0.19 | 0.0020 | 9.6 | 1.15 | 28.1 | 0.004 | 0.085 | 0.108 | 30.1 | 82.2 |
| 27/7/21 | 0.80 | 12.5 | 0.17 | 0.0013 | 9.3 | 0.85 | 28.1 | 0.016 | 0.087 | 0.163 | 29.8 | 80.9 |
| 8/9/21 | 0.63 | 18.5 | 0.25 | bdl | 14.0 | 0.76 | 37.9 | 0.010 | 0.113 | 0.139 | 39.3 | 111.0 |
| 6/10/21 | 0.64 | 20.8 | 0.30 | 0.0048 | 11.4 | 0.55 | 43.3 | 0.007 | 0.131 | 0.149 | 45.8 | 122.5 |
| 4/11/21 | 0.57 | 20.3 | 0.29 | bdl | 15.7 | 0.57 | 45.1 | 0.004 | 0.130 | bdl | 47.4 | 129.5 |
| 3/12/21 | 0.63 | 14.0 | 0.17 | bdl | 11.0 | 0.42 | 30.5 | 0.005 | 0.088 | 0.105 | 33.0 | 89.3 |
| 11/1/22 | 0.71 | 17.5 | 0.26 | bdl | 11.1 | 0.55 | 38.8 | 0.005 | 0.090 | 0.102 | 40.3 | 108.7 |
| *Northern lake margin* | | | | | | | | | | | | |
| 2/2/21 | 1.29 | 4.3 | 0.06 | 0.0006 | 3.2 | 0.44 | 10.7 | 0.006 | 0.022 | 0.121 | 6.5 | 25.4 |
| 23/2/21 | 1.50 | 3.8 | 0.06 | 0.0005 | 2.8 | 0.41 | 8.8 | 0.002 | 0.029 | 0.040 | 9.5 | 25.4 |
| 20/4/21 | 1.25 | 5.5 | 0.08 | 0.0006 | 4.1 | 0.55 | 12.2 | 0.003 | 0.039 | 0.057 | 12.9 | 35.5 |
| 6/5/21 | 1.25 | 5.7 | 0.07 | bdl | 4.2 | 0.52 | 13.2 | 0.002 | 0.046 | 0.067 | 14.1 | 37.9 |
| 26/5/21 | 1.17 | 7.3 | 0.11 | 0.0009 | 5.4 | 0.73 | 16.3 | 0.015 | 0.052 | 0.085 | 17.2 | 47.2 |
| 14/6/21 | 1.10 | 10.9 | 0.16 | 0.0013 | 8.0 | 1.03 | 23.1 | bdl | 0.079 | 0.106 | 24.0 | 67.4 |
| 13/7/21 | 0.92 | 18.4 | 0.27 | 0.0026 | 13.6 | 0.73 | 40.0 | 0.013 | 0.129 | 0.177 | 40.3 | 113.7 |
| 28/1/21 | – | bdl | bdl | bdl | bdl | bdl | 0.0037 | bdl | bdl | 0.0015 | bdl | 0.005 |
| 23/2/21 | – | 0.0011 | 0.0006 | bdl | 0.0019 | 0.0179 | 0.0046 | 0.0001 | bdl | 0.0013 | 0.0029 | 0.030 |
| 18/3/21 | – | 0.0018 | 0.0010 | bdl | 0.0035 | 0.0251 | 0.0027 | 0.0001 | bdl | 0.0026 | 0.0065 | 0.043 |
| 20/4/21 | – | 0.0016 | 0.0014 | bdl | 0.0028 | 0.0273 | 0.0035 | 0.0001 | bdl | bdl | 0.0101 | 0.047 |
| 16/5/21 | – | 0.0009 | 0.0008 | bdl | 0.0018 | 0.0255 | 0.0028 | 0.0001 | bdl | bdl | 0.0055 | 0.037 |
| 26/5/21 | – | 0.0018 | 0.0015 | bdl | 0.0022 | 0.0302 | 0.0044 | 0.0001 | 0.0004 | 0.0014 | 0.0070 | 0.049 |
| 14/6/21 | – | 0.0017 | 0.0031 | bdl | 0.0032 | 0.0336 | 0.0031 | 0.0001 | bdl | bdl | bdl | 0.045 |
| 4/11/21 | – | bdl | bdl | bdl | bdl | bdl | 0.0147 | 0.0001 | 0.0159 | bdl | 0.0728 | 0.104 |
| 3/12/21 | – | bdl | bdl | bdl | bdl | bdl | bdl | bdl | bdl | bdl | bdl | bdl |
| 11/1/22 | – | bdl | bdl | bdl | bdl | bdl | 0.0038 | bdl | 0.0011 | bdl | 0.0114 | 0.016 |


**Table A2: Hydrochemical composition of precipitation samples collected at Laguna Honda. TDS = total dissolved solids, calculated as the sum of measured cation and anion concentrations; bdl = below detection limit.**

| Sampling Date | Precipitation amount (mm) | $Na^+$ (g L$^{-1}$) | $K^+$ (g L$^{-1}$) | $NH_4^+$ (g L$^{-1}$) | $Mg^{2+}$ (g L$^{-1}$) | $Ca^{2+}$ (g L$^{-1}$) | $Cl^-$ (g L$^{-1}$) | $F^-$ (g L$^{-1}$) | $Br^-$ (g L$^{-1}$) | $NO_3^-$ (g L$^{-1}$) | $SO_4^{2-}$ (g L$^{-1}$) | TDS (g L$^{-1}$) |
|---|---|---|---|---|---|---|---|---|---|---|---|---|
| 28/1/21 | 73.8 | bdl | bdl | bdl | bdl | bdl | 0.0037 | bdl | bdl | 0.0015 | bdl | 0.005 |
| 23/2/21 | 61.0 | 0.0011 | 0.0006 | bdl | 0.0019 | 0.0179 | 0.0046 | 0.0001 | bdl | 0.0013 | 0.0029 | 0.030 |
| 18/3/21 | 14.8 | 0.0018 | 0.0010 | bdl | 0.0035 | 0.0251 | 0.0027 | 0.0001 | bdl | 0.0026 | 0.0065 | 0.043 |
| 20/4/21 | 28.2 | 0.0016 | 0.0014 | bdl | 0.0028 | 0.0273 | 0.0035 | 0.0001 | bdl | bdl | 0.0101 | 0.047 |
| 16/5/21 | 39.8 | 0.0009 | 0.0008 | bdl | 0.0018 | 0.0255 | 0.0028 | 0.0001 | bdl | bdl | 0.0055 | 0.037 |
| 26/5/21 | 1.6 | 0.0018 | 0.0015 | bdl | 0.0022 | 0.0302 | 0.0044 | 0.0001 | 0.0004 | 0.0014 | 0.0070 | 0.049 |
| 14/6/21 | 39.0 | 0.0017 | 0.0031 | bdl | 0.0032 | 0.0336 | 0.0031 | 0.0001 | bdl | bdl | bdl | 0.045 |
| 4/11/21 | 56.8 | bdl | bdl | bdl | bdl | bdl | 0.0147 | 0.0001 | 0.0159 | bdl | 0.0728 | 0.104 |
| 3/12/21 | 31.2 | bdl | bdl | bdl | bdl | bdl | bdl | bdl | bdl | bdl | bdl | bdl |
| 11/1/22 | 67.6 | bdl | bdl | bdl | bdl | bdl | 0.0038 | bdl | 0.0011 | bdl | 0.0114 | 0.016 |


**Table A3: Saturation indices for lake water samples calculated for calcite, aragonite, dolomite, gypsum, anhydrite and halite using the *Aquachem* software. If the saturation index is >0, mineral precipitation will occur (orange) and if it is <0, mineral dissolution will take place (blue).**

| Sampling Date | Water level (m) | Calcite | Aragonite | Dolomite | Gypsum | Anhydrite | Halite |
|---|---|---|---|---|---|---|---|
| *Southern lake margin* | | | | | | | |
| 28/1/21 | 1.31 | 1.40 | 1.25 | 3.92 | -0.32 | -0.56 | -3.20 |
| 23/2/21 | 1.50 | 1.34 | 1.19 | 3.75 | -0.13 | -0.37 | -3.29 |
| 18/3/21 | 1.34 | 1.60 | 1.45 | 4.32 | -0.29 | -0.52 | -3.20 |
| 20/4/21 | 1.25 | 1.60 | 1.45 | 4.36 | 0.01 | -0.22 | -2.99 |
| 6/5/21 | 1.25 | 1.39 | 1.25 | 4.06 | -0.07 | -0.28 | -3.00 |
| 26/5/21 | 1.17 | 1.38 | 1.24 | 4.02 | 0.09 | -0.13 | -2.79 |
| 14/6/21 | 1.10 | 1.23 | 1.09 | 3.83 | 0.16 | -0.01 | -2.57 |
| 13/7/21 | 0.92 | 1.59 | 1.45 | 4.50 | 0.41 | 0.24 | -2.26 |
| 27/7/21 | 0.80 | 1.48 | 1.34 | 4.38 | 0.29 | 0.11 | -2.27 |
| 8/9/21 | 0.63 | 1.46 | 1.32 | 4.57 | 0.31 | 0.12 | -1.90 |
| 6/10/21 | 0.64 | 1.41 | 1.26 | 4.45 | 0.30 | 0.10 | -1.77 |
| 4/11/21 | 0.57 | 1.55 | 1.40 | 4.83 | 0.30 | 0.10 | -1.73 |
| 3/12/21 | 0.63 | 1.50 | 1.34 | 4.62 | 0.11 | -0.11 | -2.12 |
| 11/1/22 | 0.71 | 1.60 | 1.45 | 4.74 | 0.29 | 0.08 | -1.90 |
| *Northern lake margin* | | | | | | | |
| 2/2/21 | 1.29 | 1.48 | 1.33 | 4.09 | -0.27 | -0.51 | -3.14 |
| 23/2/21 | 1.50 | 1.31 | 1.15 | 3.66 | -0.11 | -0.35 | -3.27 |
| 20/4/21 | 1.25 | 1.36 | 1.22 | 3.89 | 0.01 | -0.22 | -2.99 |
| 6/5/21 | 1.25 | 1.03 | 0.89 | 3.37 | -0.04 | -0.22 | -2.98 |
| 26/5/21 | 1.17 | 0.94 | 0.80 | 3.12 | 0.14 | -0.06 | -2.76 |
| 14/6/21 | 1.10 | 0.18 | 0.04 | 1.70 | 0.28 | 0.14 | -2.45 |
| 13/7/21 | 0.92 | 1.33 | 1.20 | 4.39 | 0.25 | 0.11 | -1.91 |


**Table A4: Hydrogen and oxygen isotope composition of lake waters sampled at Laguna Honda.**

| Sampling Date | Water level (m) | δ$^{17}$O (‰) | | δ$^{18}$O (‰) | | δ$^2$H (‰) | | $^{17}$O-excess (per meg) | | |
|---|---|---|---|---|---|---|---|---|---|---|
| | | AV | SD | AV | SD | AV | SD | AV | SD | SE |
| *Southern lake margin* | | | | | | | | | | |
| 28/1/21 | 1.31 | -1.0407 | 0.0368 | -1.9475 | 0.0452 | -26.23 | 0.19 | -12 | 22 | 8 |
| 23/2/21 | 1.50 | -0.2728 | 0.0424 | -0.5003 | 0.0691 | -15.34 | 0.12 | -9 | 25 | 10 |
| 18/3/21 | 1.34 | 0.6534 | 0.0478 | 1.2590 | 0.0876 | -6.96 | 0.40 | -11 | 21 | 9 |
| 20/4/21 | 1.25 | 1.9849 | 0.0340 | 3.8151 | 0.0707 | 4.69 | 0.42 | -28 | 11 | 4 |
| 6/5/21 | 1.25 | 2.5776 | 0.0554 | 4.9768 | 0.1078 | 10.83 | 1.04 | -47 | 16 | 7 |
| 26/5/21 | 1.17 | 3.7496 | 0.0614 | 7.1817 | 0.1035 | 19.69 | 0.38 | -36 | 20 | 8 |
| 14/6/21 | 1.10 | 4.6430 | 0.0376 | 8.8853 | 0.0792 | 27.88 | 0.23 | -38 | 21 | 8 |
| 13/7/21 | 0.92 | 6.1492 | 0.0442 | 11.7629 | 0.0606 | 39.55 | 0.21 | -44 | 24 | 10 |
| 27/7/21 | 0.80 | 6.7230 | 0.0701 | 12.8911 | 0.1655 | 43.77 | 0.44 | -62 | 24 | 11 |
| 8/9/21 | 0.63 | 7.7712 | 0.1113 | 14.9142 | 0.1965 | 50.69 | 0.52 | -75 | 11 | 4 |
| 6/10/21 | 0.64 | 7.3422 | 0.1227 | 14.0932 | 0.2402 | 47.49 | 0.63 | -74 | 18 | 7 |
| 4/11/21 | 0.57 | 5.6049 | 0.0376 | 10.7706 | 0.0655 | 34.37 | 0.19 | -67 | 21 | 8 |
| 3/12/21 | 0.63 | 5.1657 | 0.0308 | 9.9724 | 0.0532 | 31.27 | 0.33 | -87 | 23 | 9 |
| 11/1/22 | 0.71 | 4.3347 | 0.0400 | 8.3460 | 0.0689 | 24.11 | 0.16 | -63 | 20 | 8 |
| *Northern lake margin* | | | | | | | | | | |
| 2/2/21 | 1.29 | -0.8126 | 0.0400 | -1.4908 | 0.0762 | -23.63 | 0.45 | -25 | 8 | 3 |
| 23/2/21 | 1.50 | -0.2994 | 0.0639 | -0.5423 | 0.0998 | -15.10 | 0.19 | -13 | 16 | 7 |
| 20/4/21 | 1.25 | 2.2815 | 0.0366 | 4.3882 | 0.0481 | 8.06 | 0.31 | -33 | 25 | 10 |
| 6/5/21 | 1.25 | 2.5265 | 0.0748 | 4.8406 | 0.1360 | 10.01 | 0.27 | -26 | 18 | 9 |
| 26/5/21 | 1.17 | 4.6855 | 0.0694 | 8.9594 | 0.1303 | 27.30 | 0.56 | -35 | 19 | 10 |
| 14/6/21 | 1.10 | 6.0275 | 0.0815 | 11.5158 | 0.1740 | 39.71 | 0.58 | -36 | 29 | 13 |
| 13/7/21 | 0.92 | 8.8979 | 0.0686 | 17.0627 | 0.1365 | 60.44 | 0.62 | -75 | 8 | 3 |

**Table A5: Hydrogen and oxygen isotope composition of precipitation samples collected at Laguna Honda.**

| Sampling Date | Precipitation amount (mm) | $\delta^{17}O$ (‰) | | $\delta^{18}O$ (‰) | | $\delta^2H$ (‰) | | $^{17}O$-excess (per meg) | | |
|---|---|---|---|---|---|---|---|---|---|---|
| | | AV | SD | AV | SD | AV | SD | AV | SD | SE |
| 28/1/21 | 73.8 | -2.4728 | 0.0315 | -4.7305 | 0.0529 | -22.07 | 0.26 | 28 | 16 | 6 |
| 23/2/21 | 61.0 | -1.9875 | 0.0515 | -3.7811 | 0.0578 | -21.15 | 1.21 | 11 | 23 | 13 |
| 18/3/21 | 14.8 | -1.7849 | 0.0551 | -3.3981 | 0.0808 | -19.12 | 0.66 | 11 | 23 | 9 |
| 20/4/21 | 28.2 | -1.5073 | 0.0417 | -2.8723 | 0.0729 | -16.64 | 0.73 | 15 | 12 | 5 |
| 16/5/21 | 39.8 | -2.0599 | 0.0309 | -3.8734 | 0.0511 | -29.04 | 0.21 | -13 | 18 | 7 |
| 26/5/21 | 1.6 | -1.0101 | 0.0260 | -1.9179 | 0.0299 | -5.12 | 0.85 | 2 | 15 | 7 |
| 14/6/21 | 39.0 | -0.7757 | 0.0457 | -1.4565 | 0.0705 | -5.84 | 0.19 | -6 | 17 | 6 |
| 4/11/21 | 56.8 | -2.4097 | 0.0386 | -4.5662 | 0.0557 | -27.95 | 0.80 | 4 | 17 | 8 |
| 3/12/21 | 31.2 | -3.7851 | 0.0167 | -7.1806 | 0.0254 | -47.34 | 0.20 | 13 | 12 | 5 |
| 11/1/22 | 67.6 | -3.2166 | 0.0356 | -6.1295 | 0.0490 | -44.48 | 0.17 | 25 | 22 | 8 |