# Peer review of "How seasonal hydroclimate variability drives the triple oxygen and hydrogen isotope composition of small lake systems in semiarid environments"

_Hydrology and Earth System Sciences, 2024_

## Author Comment (AC1)

*Voigt and coauthors investigate the water balance and chemistry of a small lake in an arid environment with dynamic, seasonal hydrology. This is an interesting and timely subject because small arid lakes are prone to changes in the face of anthropogenic climate change and are rarely in hydrologic steady state. Voigt and coauthors characterize the hydrologic conditions in the lake with the following modeling and empirical efforts: 1) collection of water samples for isotopic (d18O, d17O, and d2H) and anion/cation analysis, 2) simulations to match the water isotopes, and 3) isotope mass balance considering lake levels, and 4) model water in the lake via satellite imagery, bathymetry, and an estimate of input water from soil moisture. The lake water balance is controlled by evaporation, precipitation, and basin discharge. There is pronounced seasonality in the water isotopes, including in triple oxygen isotopes. Evaporation and relative humidity are two main controls on the isotopic composition of the lake water.*

*This paper is an impressive combination of empirical data (isotope and environmental monitoring) and modeling approaches. ic hydrological processes in a small, arid lake. Combined, these results yield a comprehensive description of the dynam Their use of triple oxygen isotopes to identify non-steady state hydrological processes is exciting and demonstrates the value of this novel technique in applications to modern hydrology. Furthermore, I think that while it is not surprising that water isotopes vary in a small catchment throughout a year, it continues to be important to point this out to the paleoclimate community. I recommend publication of this manuscript with very minor revisions. Below I offer some comments and suggestions for improvement.*

**We thank the reviewer for his insightful comments that will help significantly to improve the manuscript. Below, we respond to each of the specific comments (blue) and indicate how the comment will be addressed in the revised manuscript (yellow).**

**Specific Comments**

*Line 135: How are you estimating measurement precision? Is this the S.D. of the seven injections for a single vial, or is it the S.D. of multiple replicates of the standards run over time? I would recommend using the latter, and also incorporating an estimate of error in your normalization scheme, to arrive at an accurate estimate of error. See Hutchings and Konecky (2023).*

**The long-term precision is based on multiple replicates of an analytical standard that was measured alongside with the samples as control. We will specify this in the manuscript as follows:**

**"The long-term precision for $\delta^{17}O$, $\delta^{18}O$, and $\delta^2H$ was 0.08‰, 0.15‰, and 0.7‰, respectively, based on the analysis of an analytical standard (n=35) conducted alongside the samples during the analysis period (April 2021 to April 2022). For $^{17}O$-excess and d-excess, the precision was 13 per meg and 1‰, respectively."**

*Line 330: the spatial variation in lake water isotopes is a very interesting finding. I would like to see this emphasized for paleoclimate applications. Paleoclimate workers often sample lacustrine sediments at a single location (one outcrop, one core). This*

*result implies that, for small lakes in the geologic record, we should be sampling horizontal transects.*

**This comment has also been raised by the second reviewer. We will add a short discussion on the implication of the spatial variations for the paleo-data in the discussion section:**

**"Spatial variations in the isotope composition of lake water should be considered when interpreting isotope data obtained from paleo-archives from lake sediments. Recently, gradients in the isotope composition of gypsum hydration water from the lake margin to the centre have been linked to increasing lake water evaporation due to decreasing water level (Cañada-Pasadas et al., 2024). Sampling lake sediments along horizontal transects can thus provide information at different evaporation stages."**

**Further, we will point out implications of our study for paleoclimate studies in the conclusions:**

**"Our results have significant implications for interpreting isotope data retrieved from lake sediment archives in paleoclimate studies. Lake evaporites, such as gypsum, capture the isotopic composition of the lake water at the time of their formation. Recently, Cañada-Pasadas et al. (2024) demonstrated that surface gypsum samples of an Andalusian wetland reflect lake water isotope conditions during spring and early summer, rather than the annual average. Determining the timing of gypsum formation is thus essential to ensure accurate paleo-interpretations. Additionally, spatial variations in the isotope composition of lake water must be considered. Lake sediment sampling along horizontal transects can reveal information on paleo-environmental conditions at different evaporation stages."**

*Line 365: this finding points to a strong need to measure triple oxygen isotopes in water vapor. You may consider highlighting this result in the conclusions.*

**Indeed, long-term isotope records of atmospheric water vapor are scarce, especially in semarid regions, where precipitation is rare, and only one continuous record of $^{17}$O-excess$_V$ has been published so far (Voigt et al., 2023). These records would help to better constrain this variable and it's seasonal variability, and reduce the model uncertainty.**

**We added the following paragraph to the conclusions:**

**"Sensitivity analyses showed that the isotope composition of atmospheric water vapor is a key parameter determining isotope fractionation of lake water during evaporation. In the absence of direct measurements, it is often estimated from precipitation data assuming isotope equilibrium. However, in semi-arid regions, the seasonality of precipitation challenges accurate estimation of the isotope composition of atmospheric water vapor. Establishing long-term records of atmospheric water vapor**

**isotopes, particularly in regions of low or highly seasonal rainfall, would help to better constrain this variable, enhancing model simulations and reducing uncertainty in lake isotope mass balance calculations."**

*Line 385: Do you have any thoughts on why the C-G evaporation model is unable to match the isotope data given the measured parameters?*

**Model-data deviations can be mostly attributed to uncertainty in the effective relative humidity, but also in the isotope composition of atmospheric water vapor and the turbulence coefficient.**
**In our model simulations, we used daily average relative humidity values from a nearby meteorological station. The local relative humidity may deviate from these observations due to 1) microclimate created by the lake environment, 2) diurnal variations in the evaporation rate or 3) moisture build-up in the atmosphere due to evaporation of lake water.**
**Uncertainty in the isotope composition of atmospheric water vapor mainly arrise from the lack of direct measurements. We estimate the isotope composition of atmospheric water vapor from monthly precipitation data, assuming equilibrium. However, these data does not account for intramonthly variability. The isotope composition of atmospheric water vapor can change rapidly on daily or sub-daily scale, e.g., due to 1) change in air mass sources, 2) rainout or 3) local mositure recycling (evapotranspiration).**
**Finally, the turbulence coefficient is an empirical parameter, whose variability is not well understood. We used a value of 0.5, which within the range of previously suggested values (Gonfiantini, 1986). However, seasonal variations may occur due to changes in wind activity.**
**All these factors are discussed in new line 430-456.**

*Line 430 - 455: this discussion text mostly answers my above question. This data highlights that diurnal fluctuations are important to semi-arid lake hydrology, perhaps more so than seasonal or annual conditions. This finding could be useful for understanding which/how anthropogenic climate changes will impact arid environments, and which parameters should be considered when examining model predictions.*

**See reply to the question above.**

*Line 444: Is there a citation showing water vapor build up above lakes during periods of high evaporation that could support this idea? This set of sentences is confusing - why does more wind correspond to lower turbulence? This is the opposite of what I would expect.*

**The turbulence coefficient n that describes the proportion of molecular diffusion on total diffusion is weighted by a factor θ, which accounts for the vertical mixing of the atmosphere. This factor usually equals unity and is therefore neglected. However, a large evaporation flux from the lake surface can lead to a vertical gradient in the atmospheric water content and leads to a decrease in θ and thus n (Gat et al., 1994). Moderately to large**

lakes have been shown to modify their boundary layer due to admixture of evaporated water into the overlying air mass (Gat et al., 1994; Vallet-Coulomb et al., 2008; Jasechko et al., 2014).
We modified this paragraph in the main text as follows:

"In contrast, in summer, the model requires a relative humidity higher than the observed value to achieve a better agreement between the modelled and the observed isotope composition of lake water. This counterintuitive observation may result from strong evaporation during summer, which increases atmospheric relative humidity. The upward evaporation flux contributes to moisture buildup and perturbs the atmospheric boundary layer. This perturbation can be accounted for in the model by weighting the turbulence coefficient with a transport parameter, θ (Gat et al., 1994; Gibson et al., 2016b). For the Great lakes region, Gat et al. (1994) found θ to be 0.88. However, values closer to 1 are expected for smaller lakes (Gibson et al., 2016b). Alternatively, the lower turbulence coefficient in summer may result from stronger atmospheric turbulence caused by a higher frequency of windy days."

*Line 468 - Could the "end of the dry season" gray dots also be explained by evaporation from a different source? The points do not quite match up with the concave up prediction from the model - they form more of a cluster, not a trend. While the concave-up/looping prediction matches with previous data (Voigt et al., 2023), the data in this paper do not strongly support that prediction. I would suggest modifying the text to describe this slight disagreement, and possible offer an explanation for the offset.*

At the end of the dry season, the lake water level has dropped to only 0.6 m so that changes in atmospheric boundary conditions, especially reative humidity, can rapidly affect the isotope composition of lake water. For example, in Figure 5, the simulated $\delta^{18}O$ (black line) decreases by more than 10 ‰ at the end of October/beginning of November 2021 only due to an increase in RH from about 50 % to close to 100% associated with a a few days of precipitation. Notably, the admixture of precipitation has only a minor impact on the isotope composition of lake water due to mass balance consideration. The variability is largely driven by changes in relative humidity. Rapid changes in relative humidity, e.g. between rainy and non-rainy days may explain the large variability of the isotope composition of lake water at the beginning of the rainy season. The dashed gray line in Figure 6 only represents exemplary the average for a three-months periods and does not capture these variations.

*Furthermore, why do you think that the January 2022 samples do not 'complete' the cycle, and end up agreeing with the Jan 2021 samples? Are the conditions (or antecedent conditions) different comparing the two Januaries?*

Indeed, the conditions are different. In January 2022, the lake water level is 0.6 m lower than at the beginning of the study period. That's why the lake water is more evaporated in terms of its isotope composition.

*Figure 5 caption: there is only a central panel, rephrase*

**Done.**

*Figure 6- It might be useful to add a gradient of color within your time blocks to demonstrate that these isotopic values are evolving towards the steady state. This might be a challenge, though, with the rainbow in the background.*

**We will add a gradient of color to the illustrated isotope data:**

[Figure]

*Table S3: Please report d17O and d18O to the third decimal point as this information is needed to calculate D17O in per meg.*

**We will modify the supplementary tables according to the journal guidelines.**

***Technical corrections***

*Line 26 (and elsewhere): Minor grammar error. The triple oxygen isotope system allows to identify non-steady state conditions --> The triple oxygen isotope system allows **us** to identify non-steady state conditions. Alternate grammatically correct structure: The triple oxygen isotope system allows the identification of non-steady state conditions.*

*Line 60: Anthropogenic climate change (delete "the")*

*Line 360: remove comma after Both*

**We thank the reviewer for noting these grammatical mistakes. We carefully revised the manuscript and corrected the respective parts.**

---

## Author Comment (AC2)

**General Comments:** *Voigt et al. present a manuscript detailing the monitoring and modeling of a lake in semiarid southern Spain. Using a multi-systems approach, they model lake levels as a function of parameters that are commonly measured in modern systems as well as characterize potential proxy variables for the reconstruction of paleo-hydrological conditions. I find their manuscript carefully written and well-presented. I have a series of minor questions and inquiries as line comments below. In addition, I have one main desire which would be the inclusion of more detail on how their salts-based and isotope-based modelling might influence paleohydrologic studies. This linkage is teased in the front of the manuscript but perhaps could be expanded on somewhat in the discussion.*

> **We thank the reviewer for his detailed comments, which will considerably help to improve the manuscript. The discussion of implications of our results for paleoclimate studies has also been suggested by reviewer 1. These will be addressed in the revised manuscript as outlined in the author response to reviewer 1. Below, we respond to each of the specific comments (blue) and indicate how the comment will be addressed in the revised manuscript (yellow).**

*Figure 1: Is the extrapolation of lake SA and volume consistent with previous lake levels? Or simply an extension based on the established topography? In panel (a), what do the numerical values inset on each lake contour indicate? I assume total lake volume at that water level?*

> **The small numbers in panel (a) indicate the water level in (m) along the contour line. This bathymetric map has been published previously by Castro et al. (2003). We digitalized this map, estimated the surface area for each lake contour line and linearly interpolated between the contour lines. The lake volume was then estimated based on the surface area of slices at discrete depth intervals of 0.01 m:**

[Figure]

*Lines 155-161: E_SA is estimated from potential evapotranspiration and lake surface area? Or is there additional treatment of evaporation-driven water losses from the lake? Aren't there considerations needed based on wind fields, surface roughness, air-water temperature differences, etc. when estimating something like E_SA? This is not my expertise, so perhaps a lake of this size and particular setting make these less of an issue, but my general understanding is that E_SA is typically non-trivial.*

We assume that evapotranspiration from the lake surface (E_SA) is equal to potential evapotranspiration multiplied by the simulated lake surface area as specified in line 161. Indeed, the estimation of lake evaporation is not a simple matter as there are a number of factors that can affect the evaporation rates, notably the climate and physiography of the water body and its surroundings. A wide variety of methods for estimating open water evaporation has been reported in the literature and used in practice, including pan evaporation, mass balance, energy budget models, among others. The fact that the simulated water level agrees well with the observed water level indicates that the potential evapotranspiration provides a reliable estimate of lake evaporation in our case. We will add this information in the result section (Line 273 ff.), as follows:

"Lake water level variations estimated from hydrological mass balance (Eq. 1), accounting for precipitation on the lake surface, evaporation from the lake surface and basin discharge, agree well with observations over the period of lake water isotope monitoring (model-data deviations are lower than 5 cm) (Fig. 2). This supports that major water sources are integrated, and the lake is disconnected from groundwater aquifers. It further indicates that potential evapotranspiration provides a reliable estimate of lake evaporation for the studied lake site."

Lines 173-184: This section also does not mention in any depth the relevance of the E_SA term. Obviously, from proper evaporation, we do not have any meaningful losses of salinity. However, depending on the seasonal strength of the wind field then we might expect some losses due to lake spray aerosols. Again, perhaps this is a non-issue due to the size of this lake and the environmental conditions, but some text on the E_SA term here could help more fully explain your methodology even if the processes I describe are completely negligible.

This is an interesting thought. Loss of lake water by aerosols would not only increase lake evaporation, but also lead to removal of salts from the lake water. However, considering the small size of the lake and in view of the good agreement of the observed and simulated water level (hydrological balance) we think that this process is of minor importance in our studied case.

Line 212-217: Can you expand some on how these sensitivity experiments were handled? As Monte Carlo-type simulations? Were distributions for variables normal, uniform, etc.? Were all variables allowed to freely vary within their uncertainty bounds or was each sensitivity experiment the variation of a single parameter?

We evaluated the sensitivity of the model results to the input parameters by varying one of them within the limits specified in section 3.5 and keeping other model input parameters constant. No Monte Carlo simulation was applied.

*Line 221-223: What is the error window of the bathymetric model? Perhaps the uncertainty of the lake bathymetry is small enough that any volume error from bathymetry (as opposed to lake level) is negligible, but given all the other considerations, perhaps this is worth including.*

**The lake bathymetric model has a resolution of 0.5 m. This is quite coarse, which means that the linearly interpolated bathymetric model can significantly deviate from the real situation. However, results of the sensitivity experiments showed that misestimation of the initial lake water level (and thus lake volume) has relatively little influence on the simulated lake water isotope composition (line 349-353). We think it's worth to keep these analysis results to justify the use of the linear bathymetric model.**

*Line 235-239: Another possible consideration is the difference between air temperature and lake temperature. Although this is a small lake and probably reasonably tracks monthly air temperature, diurnal variations in air temperature may result in larger-than-expected evaporation occurring during nighttime when air temperatures drop but the lake remains warm.*

**Our model simulations are performed on a daily timestep. As the lake investigated in our study is small and shallow, we assume that the lake water temperature is equal to that of the atmosphere on a daily scale. Diurnal variations are not accounted for in our simulation, but can contribute significant uncertainty to our simulation results. In Section 5.1, we discuss the importance of diurnal variations of the relative humidity along with the evaporation rate. Temperature differences between lake water and the atmosphere further increase this uncertainty. We modified the paragraph as follows to include this information:**

**"On the other hand, the effective relative humidity during evaporation may be lower than its daily average value as the evaporation rate is usually higher during daytime when relative humidity is lower (Gibson, 2002). Also, temperature differences between lake water and the atmosphere may influence evaporation rates and effective relative humidity. Constraining the effective relative humidity that drives the isotope composition of the evaporation flux with precision is thus challenging."**

Line 247-255: How well-constrained is *n*, really? A margin of 0.1 (as a SD of a normal distribution?) may be adequate, but I wonder if a larger range should have been considered.

**The "margins" given for each input parameter in Section 5.1 do not refer to uncertainty margins but to the value each parameter has changed individually, keeping other parameters constant, to quantify its impact on the simulation. For the turbulence coefficient n a change of 0.1 appears to be reasonable. The**

reader can think of a larger change (e.g. 0.2) that would result in a larger change (double) of the simulated results.

Line 270 / Figure 2: I wonder how many discrete rain events you have based on weather station data? Would it be useful (and legible) to include indicators for each rain event observed during the period of observation? You could perhaps limit indicators to rain events above some threshold value. This might be helpful in seeing the step-wise linkage between precipitation and lake level. I see the stepwise increases in cumulative precipitation at the top of Fig. 2. I wonder if a bar chart of each rain event (or binned 1 or 2-week cumulative values) would serve the visual explanation of this data better than a running accumulation timeseries? Just thoughts!

**Thanks for this thoughtful comment. Indeed, it was quite difficult to illustrate the impact of precipitation on the lake water level along with the much larger quantities of basin discharge and evaporation. In the revised version of the manuscript, we will add panel (e) of Figure 5 as an additional panel in Figure 2:**

[Figure]

Line 385-392 / Figure 5: This is great work and shows the strength of the sensitivity of isotope data to the important tunable parameters. I wonder exactly how the authors arrived at this particular solution and whether or not an automated solution approach (perhaps via Markov-Chain Monte Carlo) could identify whether or not multiple possible solutions exist. Further, I wonder if the variability in each parameter is well constrained by the bounds selected here. Certainly, we can imagine $n$ to vary more than between 0.4 and 0.6, but

also if we consider things like diurnal variability it may be possible to imagine that a 5% +/- bound on relative humidity is too conservative. Obviously, you must strike a middle point between unbounded variables and exact values – I think the authors have done a good job here generally but should, perhaps, include some additional reasoning on their stated bounds.

The input parameters for the simulation result illustrated by the dashed yellow line in Figure 6 were determined empirically. To achieve this, we divided the dataset into rainy and dry seasons and manually adjusted the input parameters within their uncertainty margins to improve model-data agreement. While this simulation result is not necessarily the best fit, it serves as an example of how deviations between modeled and observed data can be reduced. Quantifying the exact input values required for the best fit lies beyond the scope of this manuscript.

In response to the reviewer's comment, we performed a Monte Carlo simulation to validate our empirical approach. This simulation used uniform distributions for the input parameters (relative humidity, the isotope equilibrium coefficient for atmospheric water vapor, and the turbulence coefficient) and allowed them to vary within the uncertainty margins defined in the sensitivity experiments. To account for seasonal variations, the dataset was divided into rainy (January–May 2021 and October 2021–January 2022) and dry (June–September 2021) seasons. A total of 500 model runs were conducted, and the deviation between simulated and observed $\delta^{18}O$, $\delta^2H$, and $^{17}O$-excess of lake water was calculated for each run. The top 10 % of simulations (those with the lowest model-data deviations) were analyzed.

The Monte Carlo simulation results corroborate our empirical findings, showing similar trends in relative humidity and the turbulence coefficient. Furthermore, the analysis suggests that model-data deviations can be reduced by accounting for variability in the isotope equilibrium coefficient between atmospheric water vapor and precipitation. These findings are consistent with the results of the sensitivity experiments (see Fig. A6).

As the Monte Carlo simulation does not introduce new insights beyond those provided by the empirical approach and sensitivity experiments, we believe its inclusion in the manuscript is not necessary. However, we will modify the results section as outlined below, to clarify that this simulation is only one alternative to reduce model-data discrepancies.

[Figure]

**Figure R1: Similar to Fig. 5 of the main text, but here the dashed yellow lines show the top 10% of the simulated isotope mass balance obtained from the Monte Carlo simulation. The Monte Carlo simulation suggests better model-data agreement using Veq = 0.9 ± 0.1, n = 0.51 ± 0.06, observed RH – 3.1 ± 1.3% from January to May 2021 and from October 2021 to January 2022 and using Veq = 1.1 ± 0.1, n = 0.47 ± 0.05 and observed RH + 1.9 ± 2.5%, from June to September 2021. Analytical errors are smaller than symbol size.**

"No model-data agreement is found for a constant set of relative humidity, the turbulence coefficient and the isotope composition of atmospheric water vapor. Instead, seasonal variations in all three parameters need to be taken into account. Considering the results of the sensitivity experiments, better model-data agreement in the rainy season can achieved when using a relative humidity that is lower than the observed value and a slightly higher turbulence coefficient (Fig. 5a-b, yellow curve). In contrast, during the dry season, using a higher relative humidity value and lower turbulence coefficient is necessary to reduce the offset between the modelled and observed isotope composition of lake water (Fig. 5a-b, yellow curve). The simulated ¹⁷O-excess of lake water coincides only with observations when using a higher ¹⁷O-excess of atmospheric water vapor (33 per meg) (Fig. 5c, yellow curve)."

We will further modify the figure caption for Figure 5 as follows to clarify that this is only one possible solution:

"The dashed yellow line shows an alternative simulation that reduces the deviation between observed and simulated data. For this simulation, the dataset was divided in a rainy (January to May 2021 and October 2021 to January 2022) and a dry season (June to September 2021). Values of $V_{eq}$ = 1, n = 0.6, and observed RH – 5% were used for the rainy season and $V_{eq}$ = 1, n = 0.4 and observed RH +5 for the dry season. Analytical errors are smaller than symbol size."